# The ribosome-associated N-terminal acetyltransferase B coordinates global proteostasis and autophagy in plants by creating Ac/N-degrons

Xiaodi Gong [1], Marlena Pożoga [1], Jean-Baptiste Boyer [2], Yuxing Xue [3], Thierry Meinnel [2], Tanja Bange [3], Carmela Giglione [2], Rüdiger Hell [1,4] & Markus Wirtz [1,4] ✉

The N-terminal acetyltransferase B (NatB) acetylates ~20% of the eukaryotic proteome. However, the role of NatB-mediated N-terminal acetylation (NTA) for the regulation of the proteome fate remains unclear in eukaryotes. In this study, we demonstrate that CRISPR-Cas9-mediated deletion of NatB activity in plants results in significantly lowered global protein turnover due to decreased ubiquitin-proteasome system (UPS) activity and protein translation. Quantitative proteomics uncovers that NatB substrates are significantly enriched in the fraction of stabilized proteins in *natb* mutants. We provide direct evidence that the absent NTA of KIN11, a subunit of the autophagy-controlling energy sensor SnRK1, protects it from UPS-mediated destruction. The resulting accumulation of KIN11 is responsible for the increased resistance of *natb* mutants to energy limitation induced by prolonged darkness. Our findings establish NatB as a central regulator of UPS–autophagy interplay and highlight its role in maintaining proteome stability and enabling dynamic stress responses in plants.

N-terminal acetylation (NTA) is one of the most prevalent protein modifications in eukaryotes, catalyzed by a set of evolutionarily conserved N-terminal acetyltransferases (NATs)[1,2]. The substrate specificity of NATs is mainly determined by the first few amino acids of their N termini[3]. Among the eight identified NATs (NatA-NatH), NatA and NatB are two major NATs, associating with ribosomes to cotranslationally acetylate ~60% of the proteins in eukaryotes[1,4]. NatA acetylates proteins that are subject to removal of the initiator methionine (iMet) by methionine aminopeptidases (MetAPs) and display small amino acids (Ala, Ser, Thr, Val, Gly, and Cys) at their N-terminus. NatB acetylates proteins starting with an iMet followed by a polar or acidic amino acid (Asp, Asn, Glu, Gln) with nearly 100% efficiency[5,6]. Reduced activity of

NatA and NatB results in severe developmental defects in different organisms[5–9]. Unlike in yeast, where NatA deletion is tolerable, loss of NatA is lethal in all so far analyzed higher eukaryotes[8,10–13]. However, deletion of NatB, which is responsible for the NTA of ~20% of the proteomes in humans and plants, is lethal in *Drosophila melanogaster*[14], whereas two plant NatB T-DNA insertion mutants are viable but exhibit increased sensitivity to reductive stress[15]. This raises the question of whether the physiological role of NatB is evolutionarily conserved in multicellular eukaryotes of the green and red lineage.

NTA influences various aspects of protein behaviors, including protein subcellular localization, protein-protein interactions, protein complex formation, and protein aggregation[1]. Notably, NTA of certain

[1]Centre for Organismal Studies, Heidelberg University, Heidelberg, Germany. [2]Université Paris-Saclay, CEA, CNRS, Institute for Integrative Biology of the Cell (I2BC), Gif-sur-Yvette, France. [3]Department of Medicine II, LMU University Hospital, LMU Munich, Munich, Germany. [4]Cluster of Excellence GreenRobust, Heidelberg University, Heidelberg, Germany. ✉e-mail: markus.wirtz@cos.uni-heidelberg.de

proteins has been reported to create specific signals for degradation via a specific branch of the ubiquitin proteasome system (UPS), termed the Ac/N-degron pathway, in which the Ac/N-degron is recognized by a specific E3 ubiquitin ligase termed N-recognin[16]. Recently, NTA has been revealed to play a critical role in proteome stability and protein quality control across different species[17]. At a global level, depletion of NatA accelerates protein turnover of non-Nt-acetylated proteins via the newly defined GASTC/N-degron pathway and destabilizes the proteome in human, plants, and yeast[18–24]. NatB is not considered to be a main factor in yeast proteostasis, since no global effect on protein stability was observed in NatB-deficient yeast cells, albeit the translation rate was decreased by 50%[25]. In *Drosophila*, NatB-mediated NTA in specific organs shields key target proteins from proteasomal degradation by the ubiquitin E3 ligase UBR1, thereby controlling male germline stem cell differentiation and reproduction[26]. However, the UBR1 protein is not directly conserved in plants; instead, the degradation of proteins bearing type 1 and type 2 N-degrons is undertaken by separate plant-specific N-recognins, PRT6 and PRT1, respectively[27,28]. Therefore, the role of NatB in global proteome stability and protein quality control via the UPS in higher eukaryotes remains elusive.

Besides the UPS, autophagy is another key cellular quality control mechanism[29]. Autophagy is activated under nutrient starvation or other stress conditions to degrade protein complexes, protein aggregates, and damaged organelles, which enables recycling of nutrients for dynamic stress responses and clearance of unwanted proteins[30]. However, the connection between NTA and autophagy in eukaryotes is rarely investigated. In yeast, NatB is reported to be a positive regulator of autophagy, promoting autophagosome formation and autophagosome-vacuole fusion through the acetylation of Act1 and Vps[31]. In contrast, in human cell lines, NatB exhibits an inhibitory effect on autophagy by regulating the LKB1-AMPK-mTOR axis[32]. These contrasting findings imply that the impact of NTA on autophagy regulation is a consequence of the evolution of NatB substrates in different organisms.

Since the UPS and autophagy are the two major pathways for regulating cellular protein accumulation, the interplay of UPS and autophagy is crucial for cellular homeostasis and the fine regulation of the organismal development and stress responses[33]. Previous studies have shown that certain autophagy-related (ATG) proteins can be degraded via UPS, while proteasomal proteins themselves can be degraded via autophagy[34–39]. However, how NTA contributes to the regulation of UPS-autophagy interplay remains unclear in plants. In this study, we demonstrate that NatB regulates global protein turnover and destabilizes the plant proteome at least partially by the generation of Ac/N-degrons. Consequently, NatB substrates are substantially enriched in the fraction of accumulated proteins in NatB-deficient plants. We uncover that NatB-mediated NTA destabilizes KINASE 11 (KIN11), a subunit of the eukaryotic energy sensor SNF1-related protein kinase 1 (SnRK1) to attenuate autophagy in plants. This KIN11-mediated induction of autophagy is essential for the remarkable tolerance of NatB-deficient mutants to energy-limitation stress and also compensates for the significantly lowered UPS activity in NatB-deficient plants. In summary, we elucidate the molecular mechanisms by which NatB regulates protein stability by the UPS and autophagy to balance plant growth and nutrient stress responses in plants. These findings extend our knowledge on the physiological function and the regulatory mechanism of NTA in plants. In contrast to animals, NatB is non-essential in plants due to an unknown compensatory mechanism ensuring partial NTA of many NatB substrates in NatB-deficient mutants.

## Results

### Knockout of the NAA20 does not result in plant lethality nor abolish NTA of NatB substrates

In a previous study, we reported delayed plant development in T-DNA insertion mutants of the NatB catalytic subunit gene *NAA20* and the auxiliary subunit gene *NAA25*, both of which retained detectable levels of NatB complex transcripts and NAA25 protein[9]. To determine whether the complete loss of NatB in plants is lethal, as observed for NatB knockout mutants in animals[14], we generated NAA20 null mutants using CRISPR/Cas9. Two independent mutations within the N-terminal coding region of NAA20 introduced frameshifts and premature stop codons, abolishing translation of the catalytic subunit and thereby fully inactivating NatB (Fig. 1a, Supplementary Fig. 1a). Both CRISPR/Cas9 knockout lines, designated *naa20-cr1* and *naa20-cr2*, exhibited developmental retardation indistinguishable from the *naa20-1* T-DNA mutant, including reduced rosette leaf size and fresh weight (30–40% growth delay) (Fig. 1b–d). These results indicate that complete loss of NAA20 is detrimental but not lethal in plants. Genetic complementation of *naa20-cr1* with a NAA20–GFP fusion restored normal development (Fig. 1e, Supplementary Fig. 1b, c).

The N-acetylome was analyzed to evaluate the impact of NAA20 deletion on the NTA status of NatB substrates. N-terminomics identified 758 quantified non-redundant N-terminal peptides, of which 274 began at residue two due to iMet cleavage. Ninety-one N-termini retained the iMet, and 72 of these exhibited D/E/N/Q at position two, consistent with NatB substrate specificity (Supplementary Data 1).

In the wild type, 66 of 73 iMet-retaining proteins were fully N-acetylated. In contrast, both *naa20-cr1* and *naa20-1* mutants showed a pronounced reduction in NTA specifically for iMet-retaining peptides, while iMet-cleaved peptides (primarily acetylated by NatA) remained unaffected (Fig. 1f, g, Supplementary Fig. 1d, e). Notably, 33 iMet-retaining proteins displayed a more than 20% reduction in their Nt-acetylation in *naa20* mutants compared with wild type (Supplementary Data 2). All of these proteins displayed the canonical NatB substrate sequence (ME, MD, MN or MQ). Twenty-five of the NatB substrates were found in both *naa20* mutants; in those cases, the level of NTA was similarly decreased in both mutants.

Unexpectedly, not all NatB substrates in the mutants were unacetylated; many retained partial or full acetylation relative to wild type (Fig. 1f, g), suggesting the presence of a compensatory mechanism. Similar phenomena have been observed in vertebrates[40], potentially involving unidentified NATs or alternative known NATs acting on iMet-initiated peptides (e.g., NatC/E/F). To test the latter hypothesis, we overexpressed the catalytic subunits of NatC (NAA30), NatE (NAA50), and NatF (both long and short NAA60 isoforms). However, none of them restored NTA levels or rescued the developmental defects in the *naa20-cr1* mutant (Fig. 1h, i, Supplementary Fig. 2).

### NatB regulates protein turnover and destabilizes the plant proteome

Recently, the role of NatA in stabilizing the proteome has been revealed in various species[17]. NatB is responsible for the NTA of more approximately 20% of the plant proteome. To investigate its role in protein turnover and proteome stability, we measured the chymotrypsin-like protease activity of the proteasome in soluble protein extracts of the wild type and the *natb* mutants by monitoring the release of fluorescent AMC from the model substrate Z-Leu-Leu-Leu-AMC. We found that the proteasome activity was significantly reduced in the *natb* mutants compared with wild type (Fig. 2a). Consistent with this finding, both overall and K48-linked polyubiquitination levels showed reduced levels in the soluble protein fraction of *natb* mutants (*naa20-cr1* and *naa25-1*) compared with wild type when proteasome activity was inhibited by MG132 (Fig. 2b, c, Supplementary Fig. 3a, b). These findings imply a lower in vivo degradation rate by the UPS in *natb* mutants. Quantification of protein translation rates by feeding isotope-labeled amino acids or azidohomoalanine showed that translation rates were also reduced in the *natb* mutants (Fig. 2d, Supplementary Fig. 3c, d).

To determine whether the lowered protein turnover in *naa20-cr1* specifically leads to the accumulation of NatB substrates, we

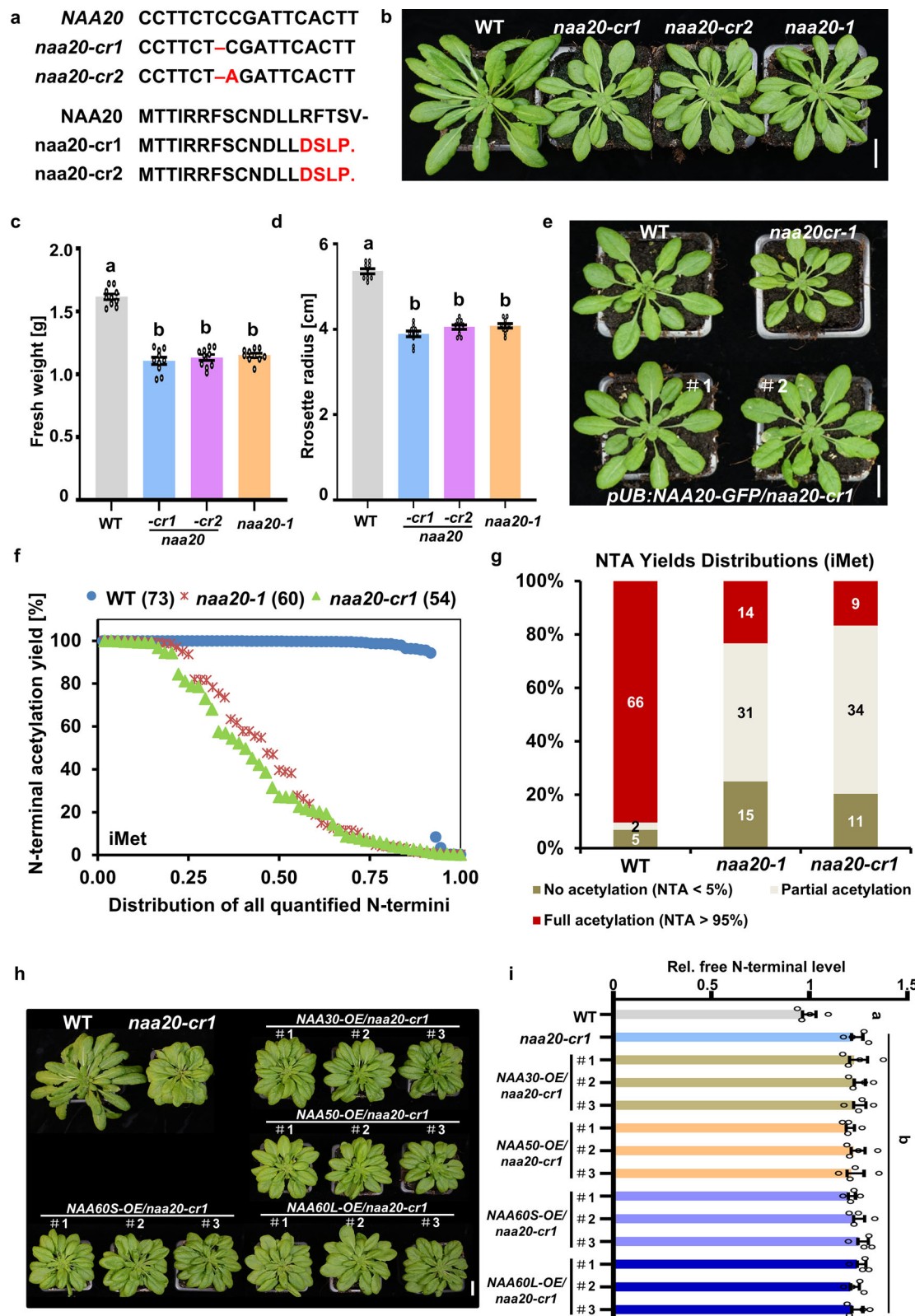

performed a quantitative proteomic analysis. A total of 3788 proteins were identified, with NatA and NatB targeting 61% and 19% of them, respectively (Fig. 2e, Supplementary Data 3). In the wild type, only 16% of stabilized proteins are NatB substrates, indicating that NatB substrates are not enriched in the stabilized protein fraction (as 19% of all identified proteins are NatB substrates). In the *natb* mutant, 30% of stabilized proteins are NatB substrates, representing a substantial enrichment of NatB substrates in this fraction when NatB is absent. The

percentage of stable NatB proteins increased from 16% in the wild type to 30% in the *natb* mutant (Fig. 2e). This finding demonstrates that NAA20 is critical for controlling the stability of its substrate in plants. A gene ontology (GO) analysis using agriGO v2.0[41] revealed that the proteins with increased abundance in the *naa20-cr1* mutant were mainly enriched in translation and amino acid metabolic process pathways (Fig. 2f), which may be explained by an unsuccessful attempt to compensate for the impaired translation rate in *naa20-cr1* (Fig. 2d).

**Fig. 1 | Knockout of NAA20 inhibits plant development and decreases NTA of canonical NatB substrates. a** Sequence of the *NAA20* gene and its encoded protein in the wild type and two CRISPR/Cas9 knockout mutants. The red color indicates the CRISPR/Cas9-induced mutations resulting in frameshifts and premature termination of the NAA20 protein in *naa20-cr1* and *naa20-cr2*. **b** Representative phenotypes of 7-week-old soil-grown wild type (WT), *naa20-cr1*, *naa20-cr2* and *naa20-1* plants. Scale bar, 2 cm. **c, d** Fresh weight (**c**) and rosette radius (**d**) of 7-week-old soil-grown wild type (WT), *naa20-cr1*, *naa20-cr2* and *naa20-1* plants. *n* = 10 plants. **e** Representative phenotypes of 6-week-old soil-grown wild type (WT), *naa20-cr1* mutant, and two independent *pUB:NAA20-GFP/naa20-cr1* transgenic lines expressing the full-length *NAA20* coding sequence (CDS) fused with GFP. Scale bar, 2 cm. **f** Distribution of NTA yields from peptides starting with an iMet (predicted substrates of NatB, NatC, and NatE) in leaves of wild type (WT) and *naa20* mutants as determined by SILProNAQ. Numbers in brackets display the

quantity of detected N termini. **g** Comparison of fully, partially, and non-acetylated protein N termini starting with an iMet in leaves of the wild type (WT) and the *naa20* mutants. Numbers indicate the quantity of characterized N termini in the respective fractions. **h** Representative phenotypes of 8-week-old soil-grown wild type (WT), *naa20-cr1*, and transgenic lines overexpressing *NAA30*, *NAA50*, *NAA60S* or *NAA60L* in *naa20-cr1*. Scale bar, 2 cm. **i** Relative quantification of free N-termini level in the soluble protein fraction of soil-grown wild type (WT), *naa20-cr1*, and *naa20-cr1* overexpressing either *NAA30*, *NAA50*, *NAA60S* or *NAA60L*. Free N-termini were stained with the dye, NBD-Cl, for quantification. *n* = 4 biologically independent samples. In (**c**, **d**, **i**), data are shown as means ± SEM. Circles indicate individual data points. Different letters indicate individual groups identified by pairwise multiple comparisons with a one-way ANOVA followed by a Tukey's test ($p < 0.05$). Source data are provided as a Source Data file.

Proteins with reduced abundance in the *naa20-cr1* mutant were mainly enriched in photosynthesis and protein synthesis pathways (Fig. 2f), reinforcing the proposed role of NAA20 in determining plant protein homeostasis and plant growth.

## NatB may not directly regulate the plant proteome by modulating the stability of individual NatB substrate

To further investigate the mechanism by which NatB regulates plant proteome stability, we performed an ubiquitin capture (Ubi capture) approach to enrich ubiquitinated proteins. The Ubi capture pulldown was followed by mass spectrometry to identify and classify differentially polyubiquitinated proteins in the wild type and *naa20-cr1*. The Ubi capture results confirmed dramatically reduced polyubiquitination levels in *naa20-cr1* compared with wild type (Fig. 3a). Consistent with this, the vast majority (310 out of 314) of significantly changed polyubiquitinated proteins identified by mass spectrometry were more abundant in the wild type than in *naa20-cr1* (Supplementary Data 4), suggesting that NatB destabilizes the plant proteome. Classification of NAT substrates for less ubiquitinated proteins in *naa20-cr1* showed that NatB substrates were not significantly enriched, with only 16% of the proteins identified as NatB substrates, while NatA substrates represented 68% of the less ubiquitinated proteins in *naa20-cr1* (Fig. 3b). GO analysis using agriGO v2.0[41] revealed that a large number of less polyubiquitinated proteins in the *naa20-cr1* mutants are involved in protein hydrolysis, translation, and response to abiotic stimuli (Fig. 3c). Among these, 86% of the identified proteins involved in the translation process are NatA substrates, whereas only 4% of them are NatB substrates (Fig. 3d). However, 42% of the identified proteins involved in the ubiquitin-dependent protein catabolic process are NatB substrates (Fig. 3e). These findings suggest that NatB may not directly regulate the plant proteome by modulating the stability of all NatB substrates. Alternatively, NatB-mediated substrates may play a role upstream of the protein homeostasis cascade.

Next, we examined the protein abundance of six NatB substrates (V-ATPase subunit, RPN12a, MAT3, PBA1, HSP101, and NBR1) in the wild type and *naa20-cr1* by immunological detection. Among these, V-ATPase subunit and RPN12a have been experimentally confirmed to have reduced NTA levels in the *naa20* mutants (Supplementary Data 2), whereas the other four candidates are predicted NatB substrates based on their sequences. The results suggested that only NBR1, a selective autophagy cargo receptor, showed higher protein abundance in *naa20-cr1* compared with wild type (Fig. 3f), implying that NatB may not directly affect the protein stability of all individual NatB substrates.

## NatB-mediated NTA destabilizes NBR1 in a UPS-dependent manner

NBR1 was not quantified by the proteomics approach (Fig. 2e). To further investigate how NatB regulates the protein abundance of NBR1, we excluded the possibility of elevated NBR1 transcripts in mutants by

RT-qPCR (Supplementary Fig. 4a). In vitro acetylation assays demonstrated that the endogenous NatB complex, immunoprecipitated from a *NAA20-GFP/naa20-cr1* line using GFP beads, catalyzes the acetylation of the NBR1 N-terminus. The N-terminally acetylated NBR1 (Ac-NBR1) and the mutagenized NBR1^E2P N-terminus served as negative controls and were not accepted as substrates by NatB (Supplementary Fig. 4b, c). Next, we applied cycloheximide (CHX) chase experiments and found that NBR1 was significantly slower degraded in *naa20-cr1* than in the wild type, implying that NatB-mediated NTA promotes degradation of soluble NBR1 (Fig. 3g). To further support this hypothesis, we examined the degradation rate of the NBR1^E2P variant, which cannot be acetylated in vivo, and found that the degradation rate of NBR1^E2P was indeed significantly reduced compared with the wild-type NBR1 (Fig. 3h). Previous studies have shown that NBR1 is a selective cargo receptor for autophagy and is degraded by autophagy in the vacuole. However, we found that treatment with MG132, an inhibitor of the proteasome, also caused its protein accumulation, and after combining Concanamycin A (ConA) to block autophagy, the amount of soluble NBR1 accumulated further (Supplementary Fig. 4d). These findings imply that the UPS regulates the amount of soluble NBR1 in an NTA-dependent manner, making it available to serve as a membrane-associated autophagy cargo receptor. To further support this conclusion, we analyzed NBR1 degradation dynamics in *naa20-cr1*, when autophagy, the UPS or both are suppressed by MG132 and/or ConA treatments. When protein synthesis was inhibited by Cycloheximide (CHX), both MG132 and ConA alone inhibited NBR1 degradation, with MG132 exhibiting a more pronounced effect in plants grown under optimal growth conditions. When MG132 and ConA were applied together, NBR1 degradation was completely inhibited, indicating that NBR1 is degraded via both the UPS and autophagy pathways in vivo (Supplementary Fig. 4e). In line with this hypothesis, we found that NBR1 was also degraded in vitro in a membrane-free system. The degradation rate of NBR1 in this membrane-free system was slower in the *naa20-cr1* and could be inhibited in the wild type by the addition of MG132 (Supplementary Fig. 4f). Taken together, these results demonstrate that, besides autophagic degradation, NBR1 undergoes UPS-mediated degradation by creating Ac/N-degron.

N-terminal acetylation can influence not only protein stability but also subcellular localization and protein–protein interactions. Next, we investigated the localization of NBR1 using stable *NBR1-mCherry-GFP/WT* and *NBR1-mCherry-GFP/naa20-cr1* transgenic lines. Microscopy results revealed that in wild-type root tip cells, NBR1 shows a cytosolic and a punctate localization pattern, which is consistent with previous studies[42]. In *naa20-cr1* mutant, the localization pattern of NBR1 remains unchanged, indicating that the absence of NTA does not affect NBR1 localization (Supplementary Fig. 4g). To investigate whether loss of N-terminal acetylation alter the interaction between NBR1 and ATG8, we performed Co-IP assay using stable *NBR1-HA/WT* and *NBR1-HA/naa20-cr1* transgenic plants. The results showed that NBR1 still interacts with ATG8 in *naa20-cr1*, suggesting that the absence of NTA

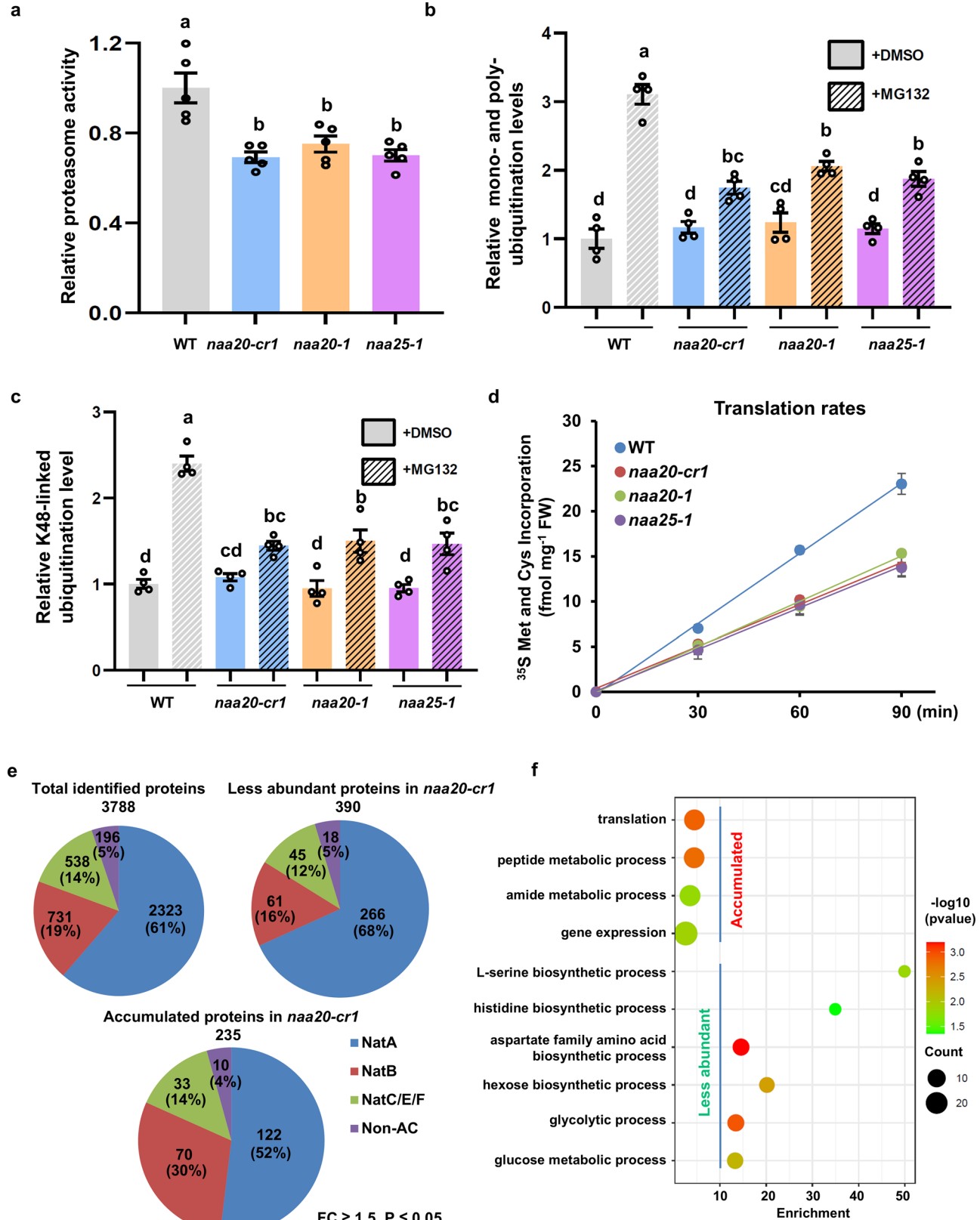

does not affect the interaction between NBR1 and ATG8 (Supplementary Fig. 4h).

## NatB-deficient plants display higher autophagic flux

Previous reports have demonstrated that a lowered proteasome abundance could be caused by enhanced autophagy[43]. To investigate whether the lower protein turnover via the UPS in the NatB mutant might be partially a consequence of a higher autophagy level in NatB-deficient mutants, we examined the K63-type polyubiquitination signals in the wild type and *naa20-cr1* plants after inhibition of autophagy by ConA. In contrast to the K48-linked polyubiquitination (Fig. 2c), the K63-linked polyubiquitination was significantly increased in *naa20-cr1*

**Fig. 2 | NatB positively regulates protein turnover in plants. a** Quantification of proteasome activity in leaves of 6-week-old wild type (WT) and *natb* mutants grown on soil under short day conditions (*n* = 5 biologically independent samples). **b, c** Quantification of global mono- and poly-ubiquitination levels with the specific FK2-ubiquitin antibody (**b**) and the global K48-linked poly-ubiquitination levels with the specific K48-ubiquitin antibody (**c**) in the wild type (WT) and *natb* mutants after treatment with or without the proteasome inhibitor MG132 for six hours (*n* = 4 biologically independent samples). **d** Translation rate in the wild type (WT) and *natb* mutants monitored by incorporation of $^{35}$S-isotope-labeled sulfur amino acids into foliar proteins of 6-week-old soil-grown plants (*n* = 3 biologically independent samples). **e** Quantitative proteomic analysis of the wild type (WT) and the *naa20-cr1* mutant. Total identified proteins, proteins more stable in wild type (WT), and proteins more stable in *naa20-cr1* were subdivided according to their recognition motifs detected by the NTA machinery in plants. Numbers indicate the quantity of detected proteins in each category. The contribution of the category to the total number in percentage is provided in brackets. Proteins with a fold change (FC) ≥ 1.5 and a two-sided Student's t-test *p*-value < 0.05 were considered as significantly different (*n* = 3 biologically independent replicates). **f** Gene Ontology (GO) enrichment analysis of the proteins that are significantly altered in their abundance in the *naa20-cr1* mutant. This analysis was performed using online tools (https://systemsbiology.cau.edu.cn/agriGOv2/index.php and https://www.bioinformatics.com.cn/). The color scale indicates different thresholds of the *p*-value, and the size of the dot indicates the number of genes corresponding to each pathway. In (**a**–**c**), data are shown as means ± SEM and all experiments were repeated three times with consistent results. Circles indicate individual data points. Different letters indicate individual groups identified by pairwise multiple comparisons with a one-way ANOVA followed by a Tukey's test (*p* < 0.05). Source data are provided as a Source Data file.

compared with wild type, indicating a higher autophagic flux in the *naa20-cr1* mutant (Fig. 4a, b). This conclusion was further supported by the detection of autophagy-related proteins ATG8 and ATG3, which are essential for autophagosome formation, in *natb* mutants and the wild type. The abundance of these ATG proteins was significantly increased in *natb* mutants (Fig. 4c–e). This increase also applied to ATG8-PE, a post-translationally lipidated proteoform of ATG8 covalently conjugated to phosphatidylethanolamine, which is essential for autophagosome membrane expansion (Fig. 4c), and the ATG5-ATG12 conjugated complex, which is essential for autophagosome formation by acting as an E3-like enzyme that promotes ATG8 lipidation (Fig. 4e). Neither ATG8–PE nor ATG5-ATG12 conjugated complex was detected in the *atg5-1* mutant (Fig. 4c, d). Real-time PCR analysis revealed no significant increase in the expression levels of these genes in the mutant, suggesting that the increase of ATG proteins may be associated with changes in post-translational modifications (Fig. 4f).

Next, we applied the autophagy marker GFP-ATG8a to analyze the autophagic flux in the wild type and the *naa20-cr1* mutant. Non-invasive confocal laser scanning microscopy of *GFP-ATG8a/naa20-cr1* and *GFP-ATG8a/*wild type demonstrated that after selective inhibition of autophagic flux by ConA application, darkness-treated *naa20-cr1* mutants accumulated significantly more autophagic bodies in their vacuole than the wild type (Fig. 4g, h). Moreover, to further substantiate the increased autophagic flux in the *naa20-cr1* mutant, a GFP-cleavage assay was performed. Darkness treatment caused a substantially stronger induction of GFP release from GFP-ATG8a in the *naa20-cr1* when compared with the wild type as indicated by the higher GFP to GFP-ATG8a ratio in *naa20-cr1* at all times of darkness treatment (4, 8, and 12 h, Fig. 4i). Taken together, these data demonstrate a higher autophagic flux in the absence of NatB.

## NatB represses autophagy in plants
The connection between N-terminal protein acetylation and autophagy remains unaddressed in plants so far. To verify whether the constitutively elevated autophagic flux could cause more efficient physiological adaptation to stress conditions in *naa20* mutants, we subjected the wild type, two *naa20* mutants, and the autophagy-deficient mutant *atg5-1* to elongated darkness stress. Both *naa20* mutants exhibited significantly higher survival rates compared with the wild type upon darkness treatment. As expected, the autophagy mutant *atg5-1* was susceptible to prolonged darkness stress (Fig. 5a, b). Consistent with this, the *natb* mutants exhibit delayed senescence after growth in short-day condition for 10 weeks and transfer to long-day condition for 2 weeks (Supplementary Fig. 5a). Furthermore, the *naa20-cr1* showed higher resistance and more fresh weight accumulation upon nitrogen and sulfur starvation conditions. The autophagy-deficient mutant performed worse than the wild type under both nutrient deficiencies. However, under control conditions, *naa20-cr1* and *atg5-1* displayed similarly decreased growth compared to the wild type (Fig. 5c, d). These results suggest that, unlike in yeast, where NatB is critical for autophagy induction[31], NatB functions as a negative regulator of autophagy in plants.

To further investigate the role of the induced autophagic flux in *natb* mutants, we crossed *naa20-cr1* with two *atg* mutants, *atg5-1* and *atg7-1*, respectively. We found that blocking autophagy did not rescue the growth defects of *naa20-cr1* under optimal growth conditions (Supplementary Fig. 5b, c). Nevertheless, both *naa20-cr1 atg5-1* and *naa20-cr1 atg7-1* double mutants lost the resistance to prolonged darkness (Fig. 5e, f). Inactivation of autophagy in the *naa20-cr1 atg* double mutants restored wild-type levels of polyubiquitination, which is in agreement with the hypothesis that induction of autophagy is partially responsible for the disturbed protein turnover in *naa20-cr1* (Fig. 5g, h). In support of a substantial crosstalk between autophagy and the UPS, the autophagy-deficient single mutants exhibited higher proteasome activity and polyubiquitination levels compared with the wild type (Fig. 5g, h, Supplementary Fig. 5d), demonstrating that the UPS is generally activated after the blockage of autophagy. To verify that crossing of *atg5-1* into *naa20-cr1* indeed inhibited autophagy in the double mutant, we crossed the GFP-ATG8a marker into this line and demonstrated the absence of autophagosome formation in this mutant (Supplementary Figs. 5e, f). Taken together, these results demonstrate that NAA20 regulates resistance to carbon starvation through autophagy and strongly suggest that constitutively induced autophagy may contribute to the downregulation of UPS in NatB-deficient plants.

## NatB-mediated NTA regulates protein stability of KIN11, a component of the energy sensor SnRK1
To understand how the darkness stress-resistant phenotype is induced in *natb* mutants, we crossed *naa20-cr1* and *nbr1-2* and found that the *naa20-cr1 nbr1-2* double mutant is still resistant to darkness (Supplementary Fig. 6a, b), implying that the accumulation of NBR1 in *naa20-cr1* does not contribute to the darkness resistance phenotype of *naa20-cr1*. We then examined the N-terminal sequences of autophagy-related proteins, as well as those related to the upstream regulatory network of autophagy induction. We found that the two catalytic subunits of the plant energy sensor SnRK1, KIN10 and KIN11, are predicted to be NatB substrates. Thus, we detected the activation status of SnRK1 and found that pKIN10 and pKIN11, two proteoforms that are more phosphorylated at the kinase activation loop, accumulated in *naa20-cr1*, demonstrating that SnRK1 activity is substantially enhanced in the *naa20-cr1* mutant (Fig. 6a, Supplementary Fig. 6c). Consistent with the activation of SnRK1, we found that the transcriptional levels of energy-related genes downstream of SnRK1 (*DIN1*, *DIN6* and *DIN10*) were induced in *natb* mutants. Remarkably, the transcription factor *bZIP63*, which is directly phosphorylated by SnRK1 and controls transcription of these *DIN* genes, is not induced, suggesting that *DIN* gene activation is caused by post-translational modification of bZIP63. At

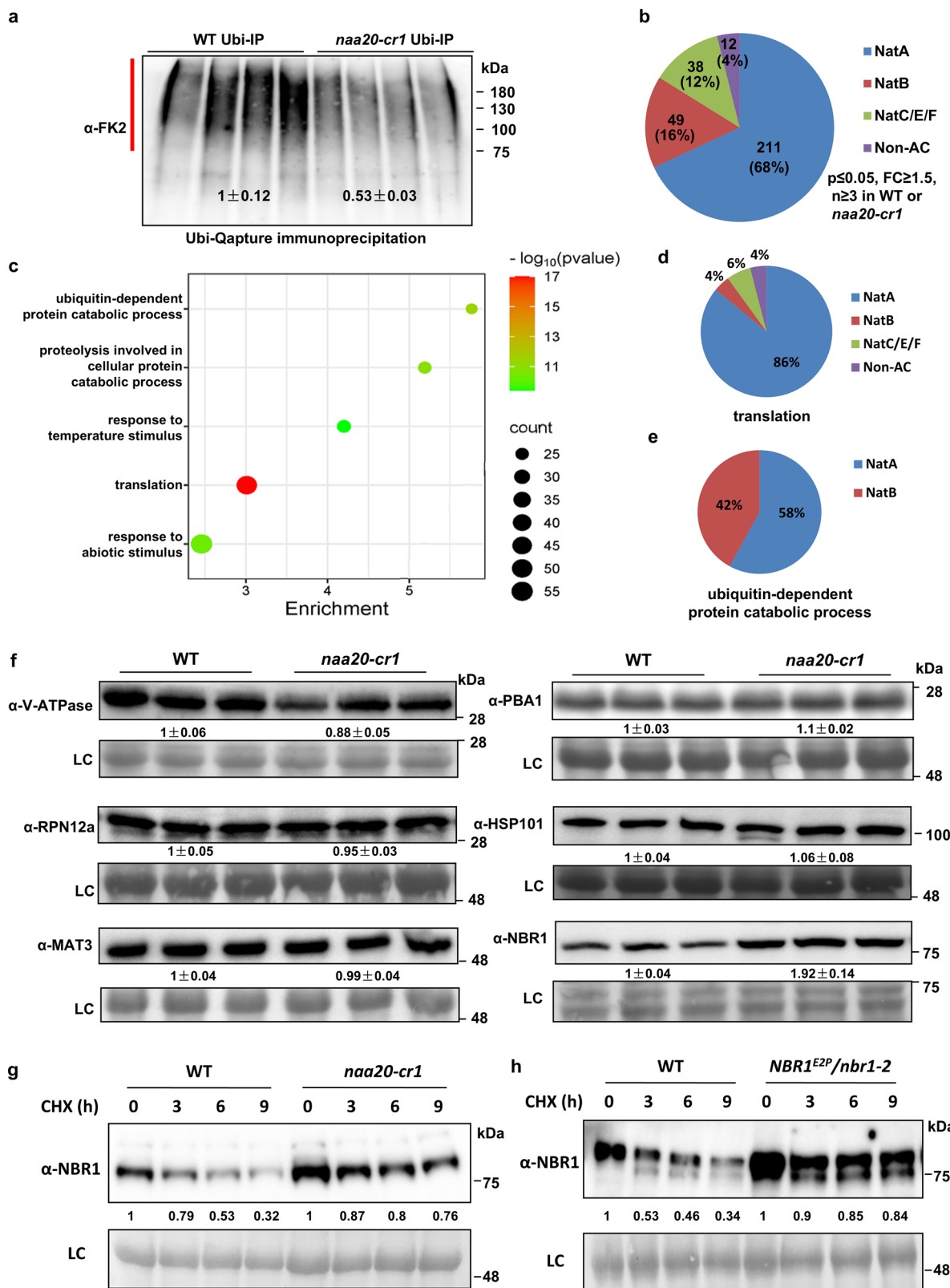

the same time, endogenous activity of the sensor kinase TARGET OF RAPAMYCIN (TOR), as determined by the ratio of phosphorylated-RPS6 to RPS6, was suppressed in *natb* mutants, which may explain the decreased translation and lowered growth of these mutants (Supplementary Fig. 6d, e). Based on these findings, we hypothesized that the absence of NTA on KIN10 and KIN11 directly causes the accumulation of pKIN10 and pKIN11, thereby enhancing SnRK1 activity in *natb*

mutants. In vitro acetylation experiments provided evidence that NatB indeed acetylates the major proteoforms of KIN10 and KIN11 (Fig. 6b). In contrast, the low-abundant proteoform of KIN10 (KIN10L), which starts with an MF, is not acetylated by NatB in vitro. Destruction of the canonical NatB substrate recognition signal in the mutagenized major proteoforms of KIN10$^{D2P}$ or KIN11$^{D2P}$ impaired acetylation of the major SnRK1 isoforms by NatB (Fig. 6b). Since we found that several NatB

**Fig. 3 | NatB destabilizes the plant proteome and impairs the stability of NatB substrates. a** Quantification of poly-ubiquitinated proteins (red line) with the specific FK2-ubiquitin antibody in the wild type (WT) and *naa20-cr1* after their selective enrichment with the Ubi-Qapture QTM matrix (*n* = 4 biologically independent replicates). Data are shown as mean ± SEM. **b** Quantitative proteomics of the selectively enriched poly-ubiquitinated proteins in the wild type (WT) and *naa20-cr1*. The pie chart depicts the classification of NAT substrates in the fraction of significantly less ubiquitinated proteins in the *naa20-cr1* mutant. Proteins with a fold change (FC) ≥ 1.5 and a two-sided Student's t-test p-value < 0.05 were considered as significantly different (*n* = 4 biologically independent replicates). **c** Gene ontology analysis of less ubiquitinated proteins in the *naa20-cr1* mutant. This analysis was performed using online tools (https://systemsbiology.cau.edu.cn/agriGOv2/index.php and https://www.bioinformatics.com.cn/). The color scale indicates different thresholds of the p-value, and the size of the dot indicates the number of genes corresponding to each pathway. **d**, **e** Pie charts representing the

less ubiquitinated proteins in *naa20-cr1* in the GO terms Translation (**d**) and Ubiquitin-dependent protein catabolic process (**e**) according to their recognition by the NAT machinery. **f** Immunoblot detection of selected NatB substrates in the wild type (WT) and the *naa20-cr1* mutant. Data represent means ± SEM (*n* = 3 biologically independent samples). Protein levels were normalized to the loading control (LC) and the detected signal was set to one in the WT. **g** Degradation rates of NBR1 in the wild type (WT) and *naa20-cr1* as determined by cycloheximide (CHX) chase experiments. Detected protein levels were normalized to the value at time point 0 h. **h** CHX chase experiments demonstrating that the degradation rate of NBR1$^{E2P}$ expressed in *pUb:NBR1$^{E2P}$/nbr1-2* transgenic plants is significantly reduced when compared with the degradation rate of wild-type NBR1. Detected protein levels were normalized to the value at time point 0 h. In (**f**–**h**), amido black staining of proteins transferred to the PVDF membrane served as internal loading control (LC) and all experiments were repeated three times with consistent results. Source data are provided as a Source Data file.

substrates accumulated in *natB* mutants (Fig. 2e), we tested the abundance of KIN10 and KIN11 in *natb* and the wild type by immunological detection. Only KIN11 but not KIN10 significantly accumulated in *naa20-cr1* (Fig. 6c, d). In this context, it is remarkable that KIN10 and KIN11 exhibit 89.2% similarity, and that only three protein domains show greater variability between the proteins (Supplementary Fig. 6f). The domain with the highest difference between both proteins is the N-terminus, suggesting that the distinct N-terminal sequences of KIN11 (MDHSSNRFGN$^{10}$-) and KIN10 (MDGSGTGSRS$^{10}$-) contribute substantially to the observed differences in accumulation.

To provide direct evidence for the specific stabilization of the non-acetylated KIN11 protein in *natb* mutants, we firstly determined the in vivo NTA levels of KIN11-HA in wild-type and *naa20-cr1* background by immunoprecipitation followed by mass spectrometry and found that the loss of NatB resulted in a two-fold lower KIN11 acetylation in *naa20-cr1* when compared to wild type (Supplementary Fig. 6g, h). We then excluded the possibility of elevated KIN11 transcripts in the *natB* mutants as the reason for KIN11 accumulation (Supplementary Fig. 6i). CHX time-chase experiment revealed that the degradation of KIN11 could be inhibited by the proteasome inhibitor MG132 (Supplementary Fig. 6j) and that the degradation rate of KIN11 was significantly lower in the *naa20-cr1* mutant than in the wild type (Fig. 6e), strongly suggesting that NTA of KIN11 leads to its degradation in a UPS-dependent manner. To further support this hypothesis, we examined the degradation rate of the KIN11$^{D2P}$ variant, which cannot be acetylated in vivo, and found that the degradation rate of KIN11$^{D2P}$-GFP was significantly decreased when compared with the regular KIN11-GFP fusion protein starting with MD (Fig. 6f). Moreover, immunoprecipitation of KIN11-GFP followed by immunological detection of polyubiquitination revealed substantially higher accumulation of polyubiquitylated KIN11-GFP in the wild-type than in the *naa20-cr1* (Fig. 6g). Furthermore, the non-acetylated KIN11$^{D2P}$-GFP was less polyubiquitinated than the acetylated KIN11-GFP in the wild type, demonstrating that the NTA status of the KIN11-GFP determines its degradation rate by the UPS (Fig. 6g). In summary, our results uncover that decreased NTA of KIN11 impairs its degradation by the UPS, which leads to the accumulation of the activated KIN11 protein. In agreement with our hypothesis that KIN10 is not destabilized in *natb* mutants due to its significantly different N-terminus when compared to KIN11, we found that KIN10 is not destabilized in *naa20-cr1* (Supplementary Fig. 6k).

### The activation of KIN11 contributes to the dark resistance of *the naa20-cr1* mutant

Previous studies have shown that despite the substantial redundancy between KIN10 and KIN11, both isoforms also have specialized functions[44,45]. Motivated by these reports, we investigated the darkness resistance of *kin10* and *kin11* mutants and found that only *kin11*, but

not *kin10*, was more sensitive to darkness treatment (Supplementary Fig. 7a, b).

To investigate whether the elevated KIN11 protein level is responsible for the increased darkness resistance of *naa20-cr1*, we crossed *naa20-cr1* with *kin10* and *kin11*. Remarkably, the darkness resistance of the *naa20-cr1 kin11* double mutant was attenuated compared with that of *naa20-cr1*. In contrast, inactivation of KIN10 in the *naa20cr-1 kin10* double mutant did not affect the darkness resistance of the NatB-deficient mutant (Fig. 7a, b). Correspondingly, two independent KIN11-overexpressing lines exhibited higher survival rates than the wild type upon darkness treatment, suggesting a role for KIN11 in regulating carbon starvation response in plants (Supplementary Fig. 7c, d). Detection of the phosphorylated SnRK1 catalytic subunits showed that the phosphorylated KIN11 was absent in the *naa20-cr1 kin11* double mutant, while the *kin11* and *naa20-cr1 kin11* mutants still showed comparable levels of phosphorylated KIN10 that were higher than those of the wild-type. These findings demonstrate that the specific activation of KIN11 is critical for the darkness resistance of *naa20-cr1* (Fig. 7c). However, the knockout of KIN11 fails to suppress the elevated ATG proteins in *naa20-cr1 kin11* double mutant, implying that activation of either KIN10 or KIN11 in the *natb* mutants is sufficient for autophagy induction under non-stressed conditions (Supplementary Fig. 7e, f). Taken together, we found that NatB regulates the interplay between UPS and autophagy in plants by fine-tuning the SnRK1 activity as well as the abundance of autophagy-associated proteins, which in turn coordinates plant growth and development and nutrient stress accordingly.

In summary, we found that the absence of NTA on many NatB substrates causes their lowered degradation by the UPS (Figs. 2c, 3g, 3h, 6e–g), which causes them to accumulate (Figs. 2e, 3f, 6c). A decreased global protein turnover and lowered UPS accompany this (Figs. 2a–d, 3a). However, we could not solely explain the decreased polyubiquitination levels in *natb* mutants by the lowered polyubiquitination of NatB substrates (Fig. 3b). Instead, we found that decreased NTA of the catalytic subunits of the energy sensor SnRK1 (KIN10 and KIN11) caused their activation (Fig. 6a), which resulted in a constitutive induction of autophagy in NatB-deficient mutants (Figs. 4, 7a), most likely contributing to compensatory general downregulation of the UPS (Figs. 2a-c, 3a, 3b). Finally, we uncovered the molecular mechanism of the darkness stress resistance of NatB-deficient mutants, which is caused by the specific accumulation of activated KIN11 protein (Figs. 5a, 5b, 5e, 5f, 7a, 7b). Our findings demonstrate that NatB controls the stability of diverse NatB substrates by the UPS and attenuates autophagy levels in plants by regulating KIN11 abundance via the generation of an Ac/N-degron (Figs. 3g, 3h, 6e–g, Supplementary Fig. 4). Consequently, NatB is a critical regulator of the UPS-autophagy interplay, which explains its resistance to diverse nutrient limitation stresses (Fig. 7d).

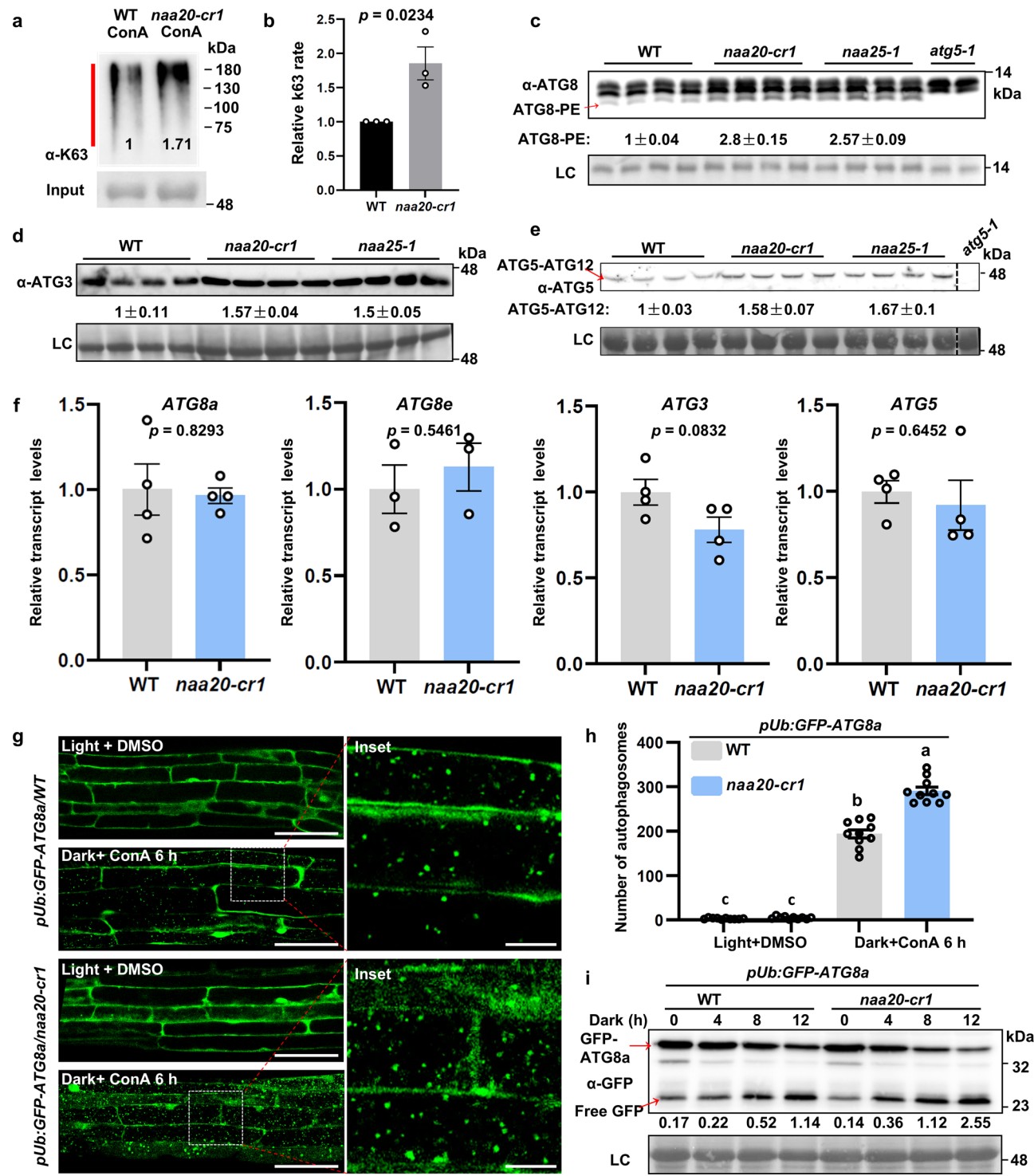

## Discussion

The complex composition and substrate specificity of NATs are evolutionarily conserved across eukaryotes, and their physiological functions are crucial for normal organismal development[1]. In multicellular eukaryotes, the absence of NatA is lethal[8,11–13]. Similarly, in *Drosophila*, deletion of NatB also results in lethality[14]. However, in plants, we reveal that deletion of the catalytic subunit of NatB, NAA20, is not lethal by identifying two NAA20 loss-of-function mutants using the CRISPR/Cas9 system. Nevertheless, this does not imply that NatB-mediated NTA is dispensable for plant viability, since the N-acetylome profiling of *naa20cr-1* revealed that the bulk of the NatB-mediated NTA in the *naa20-cr1* mutants is not entirely absent but partially decreased. In

yeast, knockout of NatB results in the almost complete absence of NTA on tested NatB substrates[25], whereas, in mammals, induced knockdown of NAA20 transcripts by 99.6% in mice and by 95% in human cell lines showed that the majority of NatB substrates remain partially acetylated[40]. These findings suggest that other unidentified NATs or alternative known NATs acting on iMet-initiated peptides (e.g., NatC/E/F) may compensate for NatB activity in higher eukaryotes. However, overexpression of the established NatC/E/F could not restore NTA levels of NatB substates in the *naa20-cr1* mutant. Given the critical function of NatB in diverse human diseases and plant growth and development, the compensatory mechanism of NatB activity will be further investigated by testing the double mutants of NatB and NatC/E/F complexes.

**Fig. 4 | The decreased UPS triggers a higher autophagic flux in the *natb* mutants. a, b** Immunoblot detection and quantization of K63-type poly-ubiquitination (red line) after enrichment with the Ubi-Qapture matrix from wild type (WT) and *naa20-cr1* treated with 1 µM ConA for 4 h. The K63-ubiquitination levels were normalized to the input and set to 1 in WT. Data represent the means ± SEM (*n* = 3 independent replicates). **c–e** Immunoblot detection of ATG8 (**c**), ATG3 (**d**) and ATG5-ATG12 complex (**e**) from wild type (WT) and *natb* mutants. In (**c, e**), *atg5-1* mutant was used as a control for antibody specificity. Data represent the means ± SEM (*n* = 4 biologically independent samples). Protein levels were normalized to the loading control (LC) and the detected signal was set to 1 in WT. **f** The transcript levels of *ATG* genes in wild type (WT) and *naa20-cr1* determined by qRT-PCR. *Actin7* (*At5g09810*) served as the reference gene. Data represent the means ± SEM (*n* = 3 for *ATG8e* or 4 for *ATG8a/3/5* biologically independent samples). **g** Confocal microscopy analysis of the autophagic bodies in cells of the root elongation zones of five-day-old *pUBQ:GFP-ATG8a/WT* and *pUBQ:GFP-ATG8a/naa20-cr1* seedlings after incubation with 1 µM ConA in the dark or with DMSO under the light for six hours. A representative picture from 10 individual seedlings is shown. Scale bar, 50 µm. Inset scale bars, 10 µm. **h** Quantitative analysis of the number of detected autophagic bodies per frame in (**g**). Data represent means ± SEM from 10 images taken from individual roots. Different letters indicate individual groups identified by pairwise multiple comparisons with a one-way ANOVA followed by a Tukey's test (p < 0.05). **i** GFP-ATG8a cleavage assay showing a higher GFP/GFP-ATG8a ratio in the five-day-old *naa20-cr1* background seedlings upon carbon-starvation treatment. The values below the immunoblot are the relative ratios of free GFP to GFP-ATG8a. In (**a, c–e, i**), amido black staining of proteins transferred to the PVDF membrane served as LC and all experiments were repeated three times with consistent results. In (**b, f**), significance was determined by the two-sided Student's t test. Source data are provided as a Source Data file.

NTA plays a crucial role in cellular processes by influencing various aspects of protein fate, including protein-protein interaction, subcellular localization, and protein folding[3]. In contrast to the previous dogma, defining NTA as a specific signal for substrate degradation, known as Ac/N-degron pathway, NatA-mediated NTA has been shown to shield the N-termini of proteins from degradation by the GASTC/N-degron pathway at a global level across yeast, plants, and humans[16,18,21]. NatB has also been shown to stabilize a handful of their substrates by protecting these proteins from degradation via the UPS in humans and plants[46–50]. However, the impact of NatB, addressing 20% of the proteome in eukaryotes[1], on the proteome stability is unclear. In Drosophila, NatB depletion has been shown to cause enhanced polyubiquitination levels in dissected testes, which was suppressed by UBR1 depletion[26]. In yeast, NatB deficiency seems not to impair turnover of proteins in a SILAC experiment, albeit it caused significantly lowered incorporation of radioactively labelled Met into the protein fraction[25].

Our study unambiguously demonstrate that NatB destabilize the plant proteome, as both global translation and global K48-polyubiquitination, which specifically targets proteins to degradation by the UPS[51], are decreased in NatB-deficient plants. Furthermore, NatB depletion causes substantial accumulation of NatB substrates. However, it would be an overinterpretation of these findings to conclude that all non-acetylated NatB substrates are destabilized in plants. The differential regulation of the NatB substrates, KIN10 and KIN11, strongly suggests that additional features of the N-terminus contribute to recognition of the acetylated N-terminus by an N-recognin. In yeast, Ac/N-degrons on NatB substrates are recognized by the E3 Ubiquitin ligase DOA10[16]. The identity of the plant Ac/N-recognin(s) remain elusive as the plant orthologues of DOA10 do not contribute to specific degradation of acetylated NatB substrates[52]. Since 6% of the *Arabidopsis* proteome is dedicated to the plant UPS[53], it is conceivable that numerous N-recognins exist that explicitly target unacetylated or acetylated subgroups of NatB substrates. Notably, in our ubiquitinome analysis, 42% of the identified proteins involved in the ubiquitin-dependent protein catabolic process are NatB substrates, which strongly suggests that NatB regulates plant proteome stability at an overall level, rather than broadly destabilizing all NatB substrates.

Due to the generation of an Ac/N-degron on KIN11, the catalytic subunit of SnRK1, NatB also influences the proteostasis of non-NatB substrates by general induction of autophagy. In humans and plants, autophagy is critical for the degradation of damaged organelles and large protein complexes, such as ribosomes and the proteasome[54]. Remarkably, human NatB also exerts control of autophagy by regulating the activity of AMPK, the human orthologue of SnRK1, via the LKB1-AMPK-mTOR axis[32], suggesting that this NatB function is conserved in multicellular eukaryotes. In our study, we found that the specific induction of autophagy was also critical for the nutrient limitation-tolerant phenotype of NatB-deficient plants, which is in full agreement with the specific induction and established role of autophagy in nutrient recycling under unfavorable environmental conditions[30]. We demonstrate that autophagy activation and darkness resistance of NatB-deficient mutants can be abolished by the loss of KIN11, suggesting that NatB serves as a critical regulatory hub in maintaining the balance between growth and nutrient recycling under suboptimal growth conditions. An impact of ribosome associated NATs on the ability of cells to adapt to environmental challenges has been also demonstrated for the NatA and NatE complexes in plants, for NatC in *Caenorhabditis elegans* and NatB in yeast[13,55–58].

The interplay of UPS and autophagy in eukaryotes is essential for the fine-tuning of cellular protein homeostasis[33]. Previous studies have shown that UPS and autophagy can mutually degrade key components of their respective pathways[34,38]. Our study uncovers that NatB creates Ac/N-degrons but also impacts proteome-wide stability via induction of autophagy, enabling it to serve as a regulator of the UPS-autophagy interplay. Moreover, given SnRK1's critical role in regulating nutrient stress responses and the TOR signaling pathway[59], NatB-mediated SnRK1 activity regulation is expected to provide further insights into the mechanisms underlying developmental plasticity in plants.

In summary, our results demonstrate that the ribosome-associated NatB complex imprints a destabilizing mark on a specific subset of its substrates. The sequence context in which the NTA of NatB substrates destabilizes is currently unknown and may depend on the recognition of diverse N-recognins. However, the absence of NatB causes a substantial decrease in global protein turnover, even though autophagy is induced in *natb* mutants. The stabilization of the NatB substrate KIN11, a catalytic subunit of SnRK1, in *natb* mutants is critical for autophagy induction and the nutrient starvation-resilient phenotype. Constitutive activation of SnRK1 might also contribute to the decreased translation and lowered growth of *natb* mutants, because of its inhibitory impact on the central growth regulator TOR, whose activity was reduced in *natb*. Our study demonstrates that, in plants the impact of NTA on the protein fate is sequence context specific. In contrast to NatA, which imprints a stabilizing mark on many housekeeping proteins, NatB destabilizes a significant subset of its substrates by acetylation.

## Methods

### Plant materials and growth conditions

*Arabidopsis thaliana* ecotype Col-0 was used as the wild type. The T-DNA insertion mutants *naa20-1* (SALK_027687) and *naa25-1* (GK-819A05) were previously published in ref. 9. The T-DNA insertion mutants *kin11* (WiscDsLox320B03) and *kin10* (GABI_579E09) were previously published in ref. 60. The T-DNA insertion mutants *nbr1-2* (GABI_246 H08), *agt5-1* (SAIL_129_B07), *atg7-1* (N39995) were obtained from the Nottingham Arabidopsis Stock Centre (www.arabidopsis.info). The *pUb:GFP-ATG8a* lines in both wild type and *agt5-1* backgrounds were previously published in ref. 61. The *pUb:GFP-ATG8a* lines

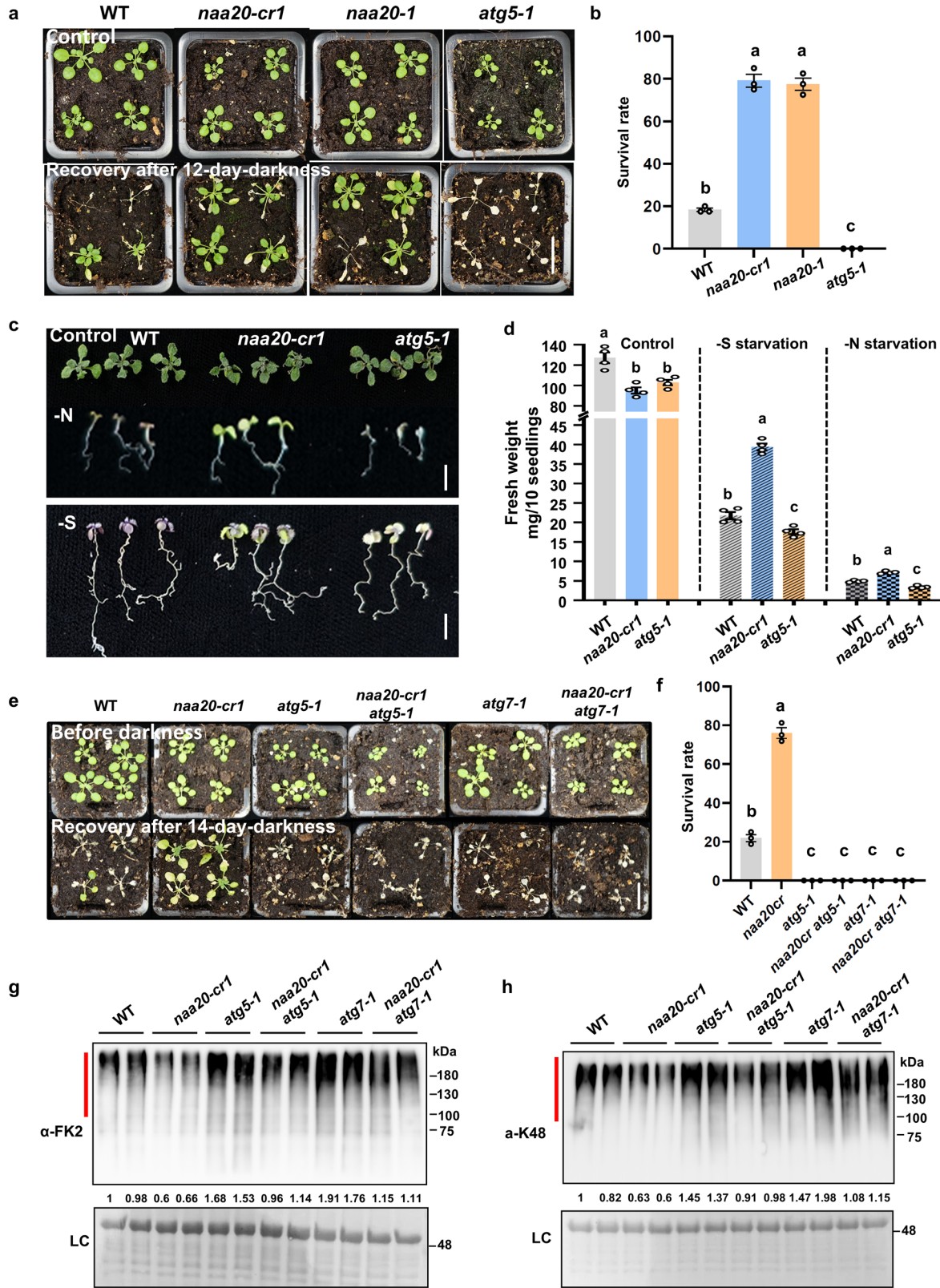

in *naa20-cr1* and *naa20-cr1 agt5-1* backgrounds were generated by genetic crossing *naa20-cr1* with the respective *pUb:GFP-ATG8a* lines. The *naa20-cr1 agt5-1*, *naa20-cr1 agt7-1*, *naa20-cr1 nbr1-2*, *naa20-cr1 kin11* and *naa20-cr1 kin10* double mutants were generated by genetic crossing the *naa20-cr1* with respective single mutants.

Seeds were sown in the pot filled with well-watered turf mixture (Ökohum, Herbertingen) supplemented with 10% [v/v] vermiculite and 2% [v/v] quartz sand. Then the seeds were vernalized in the dark at 4 °C for 2–3 days and transferred to short-day conditions (8.5 h light per day, light intensity: 100 µmol m$^{-2}$ s$^{-1}$; day temperature: 22 °C; night temperature: 18 °C; relative humidity: 50%) in plant growth chambers (ThermoTec). For seeds production or crossing, eight-week-old plants were transferred to long-day growth chambers (16-hour light per day; other conditions were the same as in short-day conditions).

**Fig. 5 | Autophagy induction in *naa20* is critical for dark resistance phenotypes. a**, **b** Representative images (**a**) and survival rates (**b**) of wild type (WT), *naa20*, and *atg5-1* plants after 12 days of carbon starvation due to elongated darkness followed by 10 days of recovery. The survival rate of 40 plants was defined as a biological replicate (*n* = 3 biologically independent replicates, 120 plants in total). Scale bar, 2 cm. **c**, **d** Representative images (**c**) and fresh weight (**d**) of wild type (WT), *naa20*, and *atg5-1* plants after 14 days of nitrogen or sulfur starvations. The fresh weight of 10 seedlings was determined and defined as one biological replicate (*n* = 4 biologically independent replicates, 40 seedlings in total). Scale bar, 0.5 cm. **e**, **f** Representative images (**e**) and survival rates (**f**) of wild type (WT), *naa20-cr1*, *atg5-1*, *atg7-1*, *naa20-cr1 atg5-1*, and *naa20-cr1 atg7-1* plants after 16 days of carbon starvation followed by 7 days of recovery. The survival rate of 40 plants was defined as a biological replicate (*n* = 3 biologically independent replicates, 120 plants in total). Scale bar, 2 cm. **g**, **h** Immunoblot detection of mono-ubiquitinated and poly-ubiquitinated proteins with the FK2 antibody (**g**) or K48-linkage-specific polyubiquitination with the K48-ubiquitin antibody (**h**) in the wild type (WT), *naa20-cr1*, *atg5-1*, *atg7-1*, *naa20-cr1 atg5-1*, and *naa20-cr1 atg7-1* plants after treatment with 50 μM MG132 for 4 h. The detected signals were normalized to the loading control (LC) and set to one in the first plant of the wild type. Amido black staining of proteins transferred to the PVDF membrane served as LC. Both experiments were repeated three times with consistent results. In (**b**, **d**, **f**), data are shown as means ± SEM. Circles indicate individual data points. Different letters indicate individual groups identified by pairwise multiple comparisons with a one-way ANOVA followed by a Tukey's test (*p* < 0.05). Source data are provided as a Source Data file.

## Constructs for genetic transformation and selection of transgenic plants

To generate *NAA20* CRISPR/Cas9 construct, single-guide RNA (sgRNA) targeting the first exon of *NAA20* was designed (Supplementary Data 5) and cloned according to the protocol[62]. Selection of NAA20 CRISPR/Cas9 seedlings was performed by applying Hygromycin B (25 mg/L) on solid ½ MS medium plates. The surviving plants were sequenced using specific primers to identify mutations (Supplementary Data 5). The CRISPR/Cas9 construct was eliminated from *naa20-cr* mutants by self-pollination and segregation.

To generate the NAA20-GFP, NBR1-HA, NBR1-mCherry-GFP, KIN11-GFP, KIN11[D2P]-GFP, KIN11-HA tagged constructs and NBR1[E2P], NAA30, NAA50, NAA60S, NAA60L overexpression constructs, the original and variant cDNA sequences of the respective genes were amplified using the PCRBIO HiFi Polymerase (PCRBIOSYSTEMS) with specific primers (Supplementary Data 5). The resulting PCR fragments were cloned into the GreenGate cloning system under the control of the Ubiquitin promoter. Final vectors were sequenced to confirm sequence identity (EUROFINs) and then stably transformed into respective plants by *A. tumefaciens*–mediated floral dipping. Selection of one-week-old soil-grown transgenic plants was performed by spraying glufosinate-ammonium solution (0.2 g/liter) three times within 6 days. The surviving plants were genotyped by PCR using specific primers or sequenced to confirm the variants and backgrounds (Supplementary Data 5).

## RNA extraction, RT-qPCR and RT-PCR

Total RNA was isolated from 100 mg of leaf materials of soil-grown plants using the Universal RNA Kit (Roboclon). Subsequently, first-strand cDNA synthesis was carried out with the Fast Gene Scriptase II cDNA Synthesis Kit (NIPPON Genetics) following the manufacturer's instructions to generate templates for both RT-qPCR and RT-PCR analyses. RT-qPCR was performed using the qPCRBIO SyGreen Mix Lo-ROX (PCR Biosystems) following the manufacturer's instructions. RT-PCR was performed using the FastGene Taq 2× Ready Mix (NIPPON Genetics). The specific primer pairs used are listed in Supplementary Data 5. The expression levels were normalized to the expression of *Actin7* (*At5g09810*) or *PP2A* (*AT1G69960*).

## Protein extraction and immunoblot detection

The soluble proteins were extracted from 100 mg of leaf materials from six-week-old soil-grown plants under short day condition with 300 μl of extraction buffer (50 mM HEPES pH 7.4, 10 mM KCl, 1 mM EDTA, 11 mM EGTA, and 10% [v/v] glycerol) supplemented with 10 mM DTT, 0.5 mM PMSF and 1 x complete protease inhibitor cocktail (Roche). For the detection of RPS6 and autophagy-related proteins, NuPage sample buffer (150 mM Tris-HCl pH 8.5, 2% LDS, 1 mM EDTA, 10% [v/v] glycerol, 0.22 mM Coomassie Blue G250, 0.166 mM Phenol Red and 50 mM DTT) was used to extract total protein. Subsequently, proteins were separated by SDS-PAGE or Tricine-SDS-PAGE (for ATG8-PE separation) and blotted onto PVDF membranes. Antibodies against

NAA25 (1:5,000)[9], NAA50 (1:5,000)[57], GFP (ChromoTek, Pabg1-100, 1:2,000 and Santa-Cruz Biotechnology, sc-9996 HRP, 1:2,000), K48-Ubi (Cell signaling, 8081, 1:5,000), K63-Ubi (Cell signaling, 5621, 1:1,000), Neutravidin-HRP (Invitrogen, 31001, 1:10,000), mono- and poly-ubiquitination (Enzo life Sciences, ENZ-ABS840HRP, 1:10,000), RPN12a (Agrisera, AS19 4268, 1:2,000), V-ATPase (Agrisera, AS07 213, 1:2,000), HSP101 (Agrisera, AS07 253, 1:2,000), SAM1-4 (Agrisera, AS16 3148, 1:2,000), PBA1 (Agrisera, AS19 4260, 1:2,000), ATG3 (Agrisera, AS19 4275, 1:3,000), ATG5 (Agrisera, AS15 3060, 1:2,000), ATG8 (Agrisera, AS14 2769, 1:2,000), RPS6 (Cell signalling, 2317, 1:2,000), p-RPS6 (1:5,000)[63], KIN10 (Agrisera, AS21 4581, 1:2,000), KIN11 (Agrisera, AS10 920, 1:1,000), Phospho-AMPKα (Thr172) Antibody (Cell signaling, 2531S, 1:2,000) or the secondary anti-rabbit IgG-HRP (Agrisera, AS09 602, 1:25,000) were diluted in 1× Tris-buffered saline (50 mM Tris, pH 7.6, 150 mM NaCl, and 0.05% [v/v] Tween 20) supplemented with 0.5% [w/v] BSA. Antibodies against NBR1 (Agrisera, AS14 2805, 1:5,000) was diluted in 1×TBST supplemented with 8% [w/v] nonfat milk. The resulting signals were quantified with the Fiji image processing application. All the uncropped blots are provided in the Source Data file.

## Protein degradation rate analysis

Leaf materials from six-week-old soil-grown plants or seedlings from ½ MS plates under short day condition were used to determine the protein degradation rate. For in vitro cell-free protein degradation assay, the soluble proteins were extracted from leaf materials by adding two volumes of extraction buffer (25 mM Tris–HCl pH 7.5, 10 mM NaCl, 10 mM MgCl2, 4 mM PMSF, 5 mM dithiothreitol, and 10 mM ATP). Supernatants were cleared by centrifugation at 20,000 g for 20 min at 4 °C, and incubated at 22 °C for indicated time. For CHX time chase assay, protein synthesis was inhibited by floating leaf discs or seedlings from ½ MS plates on ½ Hoagland medium (2.5 mM Ca(NO₃)₂, 0.5 mM MgSO₄, 2.5 mM KNO₃, 0.5 mM KH₂PO₄, 4 μM Fe-EDTA, 25 μM H₃BO₃, 2.25 μM MnCl₂, 1.9 μM ZnCl₂, 0.15 μM CuCl₂, 50 nM (NH₄)₆Mo₇O₂₄, and 0.6% (w/v) microagar pH 5.8) supplemented with 100 μM CHX (Sigma Aldrich) for indicated time under the light at 22 °C while shaking at 60 rpm/min. MG132 (50 μM) or ConA (1 μM) were applied to inhibit UPS and autophagy-mediated degradation. The collected samples were subjected to SDS-PAGE and immunoblot analysis using specific antibodies.

## Co-immunoprecipitation assay

HA magnetic beads and GFP agarose beads based immunoprecipitation was performed following the manufacturer's instructions. Proteins were extracted from transgenic plants expressing the respective HA or GFP fusion proteins with extraction buffer (50 mM Tris–HCl pH 7.5, 150 mM NaCl, 10% glycerol, 1 mM PMSF and 1 × complete protease inhibitor cocktail and 0.1% Nonidet P-40). To confirm the presence of the NatB complex in the IP products of *NAA20-GFP/naa20-cr1* plants, IP products were detected with GFP and NAA25 antibodies. To detect polyubiquitinated KIN11-GFP and KIN11[D2P]-GFP in *KIN11-GFP/WT*, *KIN11-*

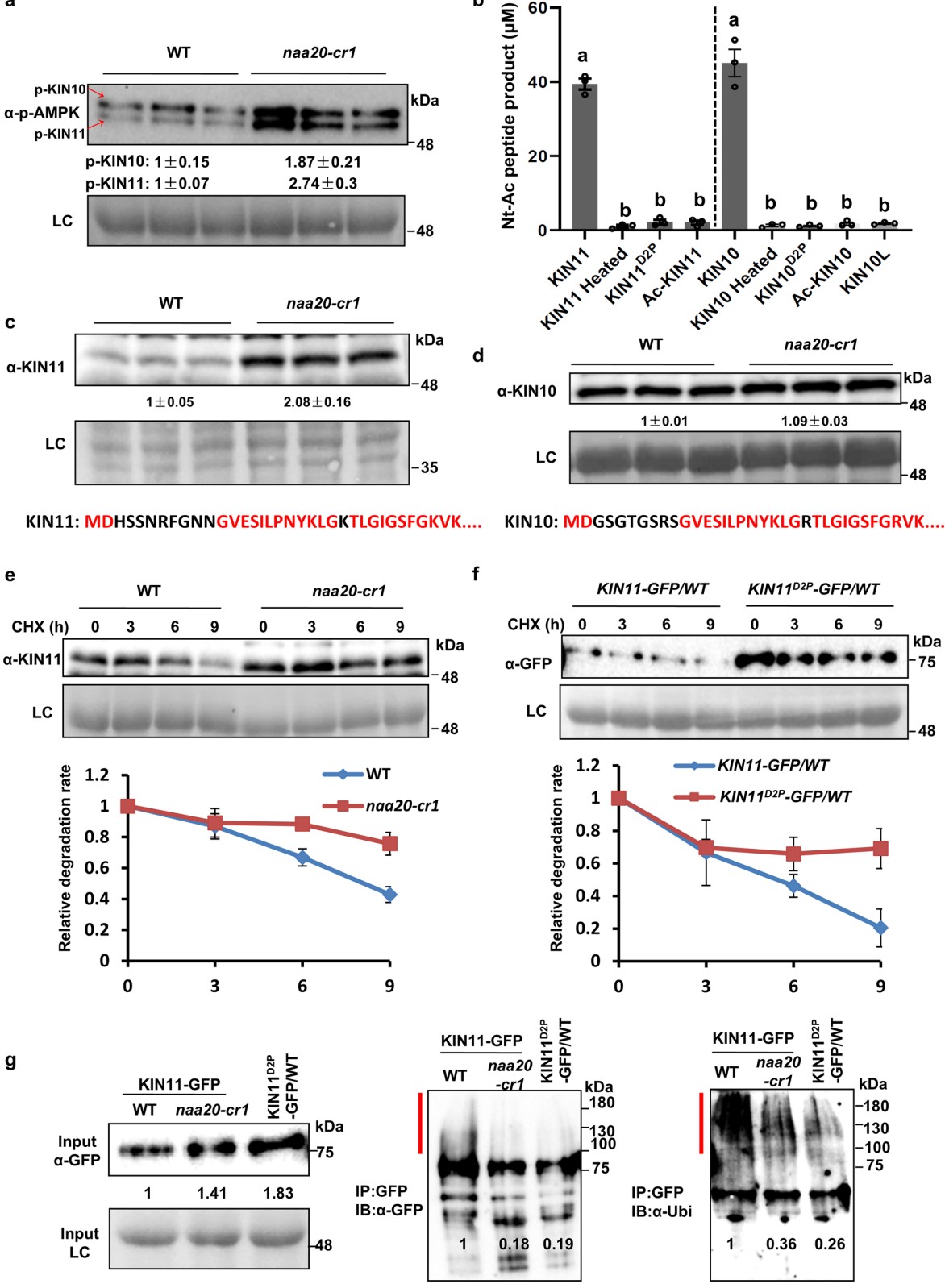

KIN11: **MD**HSSNRFGNN**GVESILPNYKLG**K**TLGIGSFGKVK....**

KIN10: **MD**GSGTGSRS**GVESILPNYKLG**R**TLGIGSFGRVK....**

GFP/naa2O-cr1 and KIN11^{D2P}-GFP/WT plants, 50 μM of MG132 was applied to leaf discs and incubated for 5 h to accumulate poly-ubiquitination. The respective IP products were detected with GFP and ubiquitin antibodies. To investigate the interaction between NBR1 and ATG8, stable NBR1-HA/WT and NBR1-HA/naa2O-cr1 transgenic plants were used.

**Determination of in vivo N-terminal acetylation of KIN11 by mass spectrometry**

SDS-PAGE gels containing HA-immunoprecipitated products from KIN11-HA/WT and KIN11-HA/naa2O-cr1 were manually excised after coomassie staining. The in-gel Trypsin digestion was performed following by Nanoflow LC-MS2 analysis using the Vanquish Neo coupled

**Fig. 6 | NatB imprints an Ac/N-degron on KIN11 to control accumulation and activity of SnRK1. a** Immunoblot detection of phosphorylated KIN10 (p-KIN10) and KIN11 (p-KIN11) in wild type (WT) and *naa20-cr1* plants. **b** DTNB-based in vitro Nt-acetylation assay confirms KIN10 and KIN11 as NatB substrates. Immunoprecipitated NatB acetylated the N-termini of KIN10 and KIN11 but not the N-terminally acetylated KIN10 (Ac-KIN10) or KIN11 (Ac-KIN11) peptides. The heated NatB complex served as a negative control. The KIN10^D2P and KIN11^D2P variants and the long KIN10 proteoform (KIN10L) were not accepted as substrates by NatB. Different letters indicate individual groups identified by pairwise multiple comparisons with a one-way ANOVA followed by a Tukey's test ($p < 0.05$). **c, d** Immunoblot detection of KIN11 (**c**) and KIN10 (**d**) in wild type (WT) and *naa20-cr1*. The N-terminal sequences of KIN11 and KIN10 are listed below. **e** Degradation rate of KIN11 in the wild type (WT) and *naa20-cr1* plants determined by immunoblot detection of KIN11 at indicated time points after 100 µM CHX treatment. **f** Degradation rate of KIN11-GFP in the *KIN11-GFP/WT* and *KIN11^D2P-GFP/WT* transgenic plants determined by

immunoblot detection of KIN11-GFP at indicated time points after 100 µM CHX treatment. **g** Detection of poly-ubiquitinated KIN11-GFP in the *KIN11-GFP/WT*, *KIN11^D2P-GFP/WT* and *KIN11-GFP/naa20-cr1* transgenic plants. The left panel shows the accumulation of KIN11-GFP in *naa20-cr-1* and the accumulation of KIN11^D2P-GFP in wild type when compared to the KIN11-GFP expression in wild type. The middle and right panel show substantially higher poly-ubiquitination (red lines) of immunoprecipitated KIN11-GFP in the wild when compared to KIN11-GFP in *naa20-cr1* or KIN11^D2P-GFP in the wild type. Numbers represent the detected signal intensities with GFP or ubiquitin antibodies. In (**a, c–g**), amido black staining of proteins transferred to the PVDF membrane served as internal loading control (LC) and all experiments were repeated three times with consistent results. In (**a, c, d**), protein levels were normalized to the loading control (LC) and the detected signal was set to 1 in the WT. Data are shown as means ± SEM ($n = 3$ biologically independent samples in (**a–d**) and $n = 3$ biologically independent replicates in **e, f**). Source data are provided as a Source Data file.

to an Orbitrap Eclipse (Thermo Fisher Scientific). An in-house packed analytical column (75 µm x 200 mm, 1.9 µm ReprosilPur-AQ 120 C18 material was used. The fragment ion mass tolerance was set to 0.02 Da, and the parent ion mass tolerance to 10 ppm. The following variable modifications were allowed: Oxidation (O), Met-loss (M), Acetyl (N-terminus), Deamidation (N, Q), whereas Carbamidomethylation (C) was set as a fixed modification. Peptide quantification was done using a precursor ion quantifier node with the Summed Intensity method set for protein abundance calculation.

### DTNB-based N-terminal acetylation assay
The in vitro NTA assay was performed as previously described in ref. 64. The purified NatB complex using GFP agarose-beads from *NAA20-GFP/naa20-cr1* plants was used. Samples without NatB complex or with heat-inactivated NatB complex were used for negative controls. The absorbance of negative controls without NatB complex was averaged and subtracted from the absorbance of the samples. The product concentration was calculated using the Lambert Beer equation and the extinction coefficient of TNB2-(13700/M cm). The following synthetic peptides (GenScript, >95% purity) were used: NBR1: [H]MESTANALVVRVSYGGVLRR[OH]; NBR1^E2P: [H]MPSTANALVVRVSYGGVLRR[OH]; Ac-NBR1: ac-MESTANALVVRVSYGGVLRR[OH]; KIN11: [H]MDHSSNRFGNNGVESILPNY[OH]; KIN11^D2P: [H]MPHSSNRFGNNGVESILPNY[OH]; ac-KIN11: ac-MDHSSNRFGNNGVESILPNY[OH]; KIN10S: [H]MDGSGTGSRSGVESILPNY[OH]; KIN10^D2P: [H]MPGSGTGSRSGVESILPNY[OH]; ac-KIN10S: ac-MDGSGTGSRSGVESILPNY[OH] and KIN10L: [H]MFRRVDEFNLVSSTIDHRI[OH]. The lysine residue in NBR1 peptides was replaced by arginine residues to minimize any potential interference by Nε-terminal acetylation.

### Determination of free N-termini
Free N-termini groups in foliar proteins from six-week-old soil-grown plants were labeled with 0.5 mM NBD-Cl (4-Chloro-7-nitrobenzofurazan, Sigma-Aldrich) as previously described in ref. 21. Prior to labeling, free amino acids and other low-molecular-weight compounds were removed using a PD SpinTrap G-25 column (Cytiva).

### Determination of proteasome activity
Proteasome activity was determined in leaves of six-week-old soil-grown plants with the proteasome substrate I (Z-Leu-Leu-Leu-AMC, Sigma-Aldrich) as previously described in ref. 22.

### Long-term carbon, nitrogen and sulfur starvation assays
For the darkness treatment and survival rate analysis, plants were germinated on soil and grown under short-day conditions for two weeks. After that, every fourth plant was picked to a pot filled with soil. After one week of recovery growth, plants were moved to a chamber set to night conditions (same conditions as in the short-day chamber,

but no light) for 8–14 days. Then the plants were moved back to the short-day conditions for 7–14 days for recovery. The survival rate was calculated by counting the plants with clearly visible intact green meristems and new leaves.

For nitrogen and sulfur starvation assays, Arabidopsis seeds were surface sterilized and seeded on solid ½ Hoagland medium plates (full, -N and -S). Plates were then kept at 4 °C in dark for two days before being moved to LD conditions. After two weeks of growth, the fresh weight of every tenth seedling was weighed. Four replicates were calculated for each phenotype or condition.

### GFP-ATG8a cleavage assay
The GFP-AtATG8a cleavage assay was performed as described previously[65]. Briefly, five-day-old seedlings grown on ½ MS medium plates containing 1% sucrose under long day conditions were exposed to continuous light for 1 day before being subjected to liquid ½ MS-C medium and kept in dark for 0, 4, 8, and 12 h. Then, the seedlings were harvested to isolate total proteins using NuPAGE buffer for immunoblotting analysis with anti-GFP antibody (Santa-Cruz Biotechnology, sc-9996 HRP).

### Quantification of global translation
Protein translation rates in leaves of six-week-old soil-grown plants were assessed by monitoring the incorporation of $^{35}$S radiolabeled cysteine and methionine using the EasyTag™ EXPRESS$^{35}$S Protein Labeling Mix (7 mCi, PerkinElmer, Waltham). In vivo protein labeling and quantification of incorporated radioactivity were performed as previously described in ref. 21.

Labeling and detection of newly synthesized proteins via L-azidohomoalanine incorporation was performed as previously described[21]. In brief, leaf discs from six-week-old soil-grown wild type and *natb* mutant were incubated for 3 h in ½ Hoagland medium supplemented with 50 µM azidohomoalanine. The incorporated azidohomoalanine was conjugated to Biotin Azide (Invitrogen) using the Click-iT™ Protein Reaction Buffer Kit (Invitrogen), and labeled proteins were detected with Neutravidin-HRP (1:100,000 dilution, Invitrogen, 31001). DMSO-treated wild-type leaf discs served as negative controls to distinguish endogenous biotinylated proteins.

### Enrichment of polyubiquitinated proteins with UbiQapture-Q matrix and quantification of ubiquitome
Enrichment of all type- and K48-polyubiquitinated proteins was performed using UbiQapture-Q matrix kit (Enzo life Sciences) as previously described in ref. 21. Proteasome and deubiquitination activity were inhibited in the leaves of wild type and *natb* plants by floating leaf discs on ½ Hoagland medium supplemented with 50 µM of MG132 (Santa-Cruz Biotechnology) and 20 µM deubiquitinase inhibitor PR-619 (Sigma Aldrich) for 6 h. To quantify the K63-polyubiquitinated

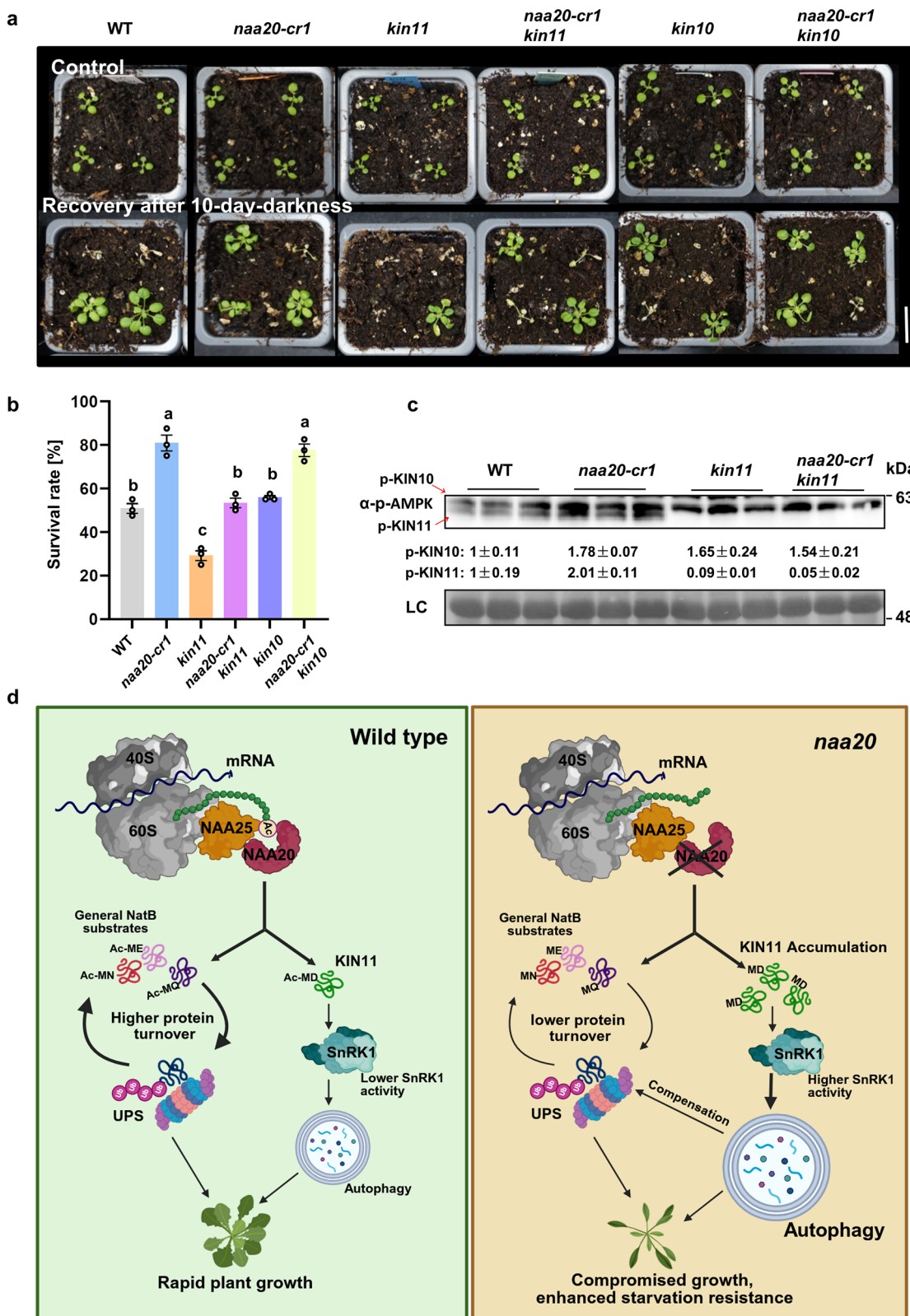

proteins, 1 μM ConA (Santa-Cruz Biotechnology) instead of 50 μM MG132 was applied to the ½ Hoagland medium to inhibit autophagy pathway.

Ubiquitome were analyzed as previously described in ref. 21. LFQ values were calculated by MaxQuant and used for quantitative data analysis. Heatmap was plotted by https://www.bioinformatics.com.cn (last accessed on 1st April 2025), an online platform for data analysis and visualization.

### LC-MS/MS measurement of whole proteomes

Proteins were extracted from dried leaves of 6-week-old soil grown plants using lysis buffer containing 75 mM HEPES pH 7.5, 150 mM KCl, 1.5 mM EGTA pH 8.0, 1.5 mM MgCl$_2$, 10% (v/v) glycerol, 0.075% NP-40 supplemented with 1 mM DTT, PhosSTOP phosphatase inhibitors, and Protease Inhibitor Mix. Proteins were precipitated in acetone and resolved in 8 M urea following a standard in-solution digest protocol as above. Peptides were separated on an EASY-nLC 1200 HPLC system

**Fig. 7 | NatB regulates stress resilience and the interplay between UPS and autophagy in plants by labeling the autophagy-inducing sensor kinase SnRK1 for degradation by the Ac/N-degron pathway. a, b** Representative images (**a**) and survival rates (**b**) of wild type (WT), *naa20-cr1*, *kin11*, *kin10*, *naa20-cr1 kin11* and *naa20-cr1 kin10* plants after 10 days of carbon starvation and 10 days of recovery. The survival rate of 40 plants grown in 10 pots were defined as a biological replicate (*n* = 3 biologically independent replicates, 120 plants in total). Data are shown as means ± SEM. Circles indicate individual data points. Different letters indicate individual groups identified by pairwise multiple comparisons with a one-way ANOVA followed by a Tukey's test (*p* < 0.05). Scale bar, 2 cm. **c** Immunoblot detection of phosphorylated KIN10 (p-KIN10) and KIN11 (p-KIN11) in wild type (WT), *naa20-cr1*, *kin11* and *naa20-cr1 kin11* plants. Data represent the means ± SEM (*n* = 3 biologically independent samples). Protein levels were normalized to the loading control (LC) and the detected signal was set to one in the WT. Amido black staining of proteins transferred to the PVDF membrane served as LC. This experiment was repeated three times with consistent results. **d** Model describing how NatB-mediated N-terminal acetylation of KIN11 controls the UPS-autophagy interplay to balance growth, global protein turnover and nutrient recycling by autophagy. Please note that lowered NTA of diverse NatB substrates contribute to the enhanced protein turnover in the wild type. This mechanism ensures rapid recycling of important and potentially highly modified proteins, like KIN11, to maintain disposability of their unmodified state for sensing purposes. Schematic created in BioRender. Wirtz, M. (2026) https://BioRender.com/5cg0wwf. Source data are provided as a Source Data file.

(Thermo Fisher Scientific, Waltham, MA, USA) using a 110 min gradient from 5–60% acetonitrile with 0.1% formic acid and then directly sprayed in an Orbitrap HF-X Mass Spectrometer (ThermoFisher Scientific, Waltham, MA, USA). The mass spectrometer was operated in a data-independent acquisition mode. The MS1 scans were aquired in a mass range of m/z 350–1000, with an Orbitrap resolution of 120,000 and a normalised AGC target of 3e6. MS2 scans were performed at a resolution of 15,000 and a normalised AGC target of 1e6. 100 windows with an isolation width of 5.6 Th (data points per peak = 6) were aquired per cycle. MS raw files were searched against the UniProt FASTA protein database of *Arabidopsis thaliana* (release April 2023). Proteins quantified in at least 3 out of 4 replicates in either WT or *naa20-cr1* mutant were filtered out, $\log_2$-transformed and imputed. Proteins with a fold change (FC) ≥ 1.5 and a Student's t-test *p*-value < 0.05 were considered as significantly different. Heatmap was plotted by https://www.bioinformatics.com.cn (last accessed on 1st April 2025), an online platform for data analysis and visualization.

### Quantification of the N-terminal acetylome
The N-acetylome was analyzed using the SILProNAQ method on proteins extracted from leaves of six-week-old soil-grown plants[66]. In brief, tissues were frozen, ground, and extracted in buffer (50 mM HEPES pH 7.2, 1.5 mM $MgCl_2$, 1 mM EGTA, 10% glycerol, 1% Triton X-100, 2 mM PMSF, 150 mM NaCl, and protease inhibitors cocktail). Free N-termini were labeled with $d_3$-acetyl groups, proteins digested with trypsin, and peptides fractionated by SCX chromatography. Enriched N-terminal peptides were analyzed by LC-MS/MS using an LTQ-Orbitrap Velos, and data were processed with Mascot Distiller and the EnCOUNTer tool[67].

### Seedling treatment and confocal microscopic analysis
Seeds of *GFP-ATG8a* in WT, *naa20-cr1*, *atg5-1*, *naa20-cr1 atg5-1* backgrounds were surface sterilized and seeded on ½ MS medium plates (with 1% sucrose). Plates were then kept at 4 °C in dark for two days before being moved to LD conditions for one week. Subsequently, plates were transferred to continuous light for one day and then immersed in liquid ½ MS medium (without 1% sucrose) with 1 μM ConA for 6 h in the dark. Seedlings treated with 1 μM DMSO under light were used as control. Roots cells in the elongation zone were imaged using the Leica Stellaris 8, HC PL APO CS2, 20x water-immersed objective and software LAS X 4.5.0.25531 (Leica, Wetzlar). The microscope was set to Pinhole 1 AU, Zoom 1.5, as well as detection with bidirectional scan with phase x30, speed 200 Hz and line average 4, line accumulation 1. The laser intensity was between 5 and 15%. Excitation at 489 nm, GFP fluorescence was recorder with HyD S1 detector at EM = 498–540 nm. Autophagic puncta labeled by GFP-AtATG8a in confocal images were identified using the ImageJ. In brief, images were background-subtracted with 25 pixels of rolling ball radius. Each image was subsequently thresholded using the MaxEntropy method and was converted to an 8-bit grayscale image. Threshold values were adjusted according to the puncta signals in original confocal images. The number of puncta in thresholded images was counted by the Analyze Particles function in ImageJ. For all puncta quantification, puncta with the size between 0.10 and 5.00 μm² were counted.

The root cells of stable transgenic *NBR1-mCherry-GFP/WT* and *NBR1-mCherry-GFP/naa20-cr1* seedlings from ½ MS plates were used to detect the subcellular localization of NBR1 using the Leica Stellaris 8, HC PL APO CS2, 40x water-immersed objective and software LAS X 4.5.0.25531 (Leica, Wetzlar).

### Quantification and statistical analysis
Data are presented as mean ± SEM. The number of replicates and experimental details are provided in the respective figure legends. Statistical analyses were performed using GraphPad Prism 9. For comparisons between two groups, an unpaired two-tailed Student's t-test was applied. Statistically significant differences a indicated with by asterisks (*$p$ < 0.05, **$p$ < 0.01). For analyses involving more than two groups, one-way ANOVA followed by a Tukey´s test was used for comparisons.

### Accession numbers
Sequence data from this article can be found in The Arabidopsis Information Resource (www.arabidopsis.org) under the following accession numbers: AT1G03150 (NAA20), AT5G58450 (NAA25), AT4G24690 (NBR1), AT5G61500 (ATG3), AT5G17290 (ATG5), AT5G45900 (ATG7), AT4G21980 (ATG8a), AT2G45170 (ATG8e), AT2G38130 (NAA30), AT5G11340 (NAA50), AT5G16800 (NAA60), AT1G64520 (RPN12a), AT4G11150 (V-ATPase), AT2G36880 (MAT3), AT1G74310 (HSP101), AT4G31300 (PBA1), AT5G09810 (ACT7), AT1G69960 (PP2A), AT3G01090 (KIN10) and AT3G29160 (KIN11), AT4G35770 (DIN1), AT3G47340 (DIN6), AT5G20250 (DIN10), AT5G28770 (bZIP63).

### Reporting summary
Further information on research design is available in the Nature Portfolio Reporting Summary linked to this article.

## Data availability
The mass spectrometry proteomics data associated to N-terminomics of *naa20* mutant lines have been deposited to the ProteomeXchange Consortium via the PRIDE partner repository with the dataset identifier PXD066411. The mass spectrometry proteomics data associated to proteomic changes of *naa20-cr1* mutant have been deposited to the ProteomeXchange Consortium via the PRIDE partner repository with the dataset identifier PXD066676. The mass spectrometry proteomics data associated to determination of polyubiquitinated proteins enriched with UbiQaputre-Q matrix in the wild-type and *naa20-cr1* mutant have been deposited to the ProteomeXchange Consortium via the PRIDE partner repository with the dataset identifier PXD054814. Source data are provided with this paper.

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

## Acknowledgements

Research at the Wirtz lab was funded by the Deutsche Forschungsgemeinschaft (DFG, German Research Foundation, project numbers 496871662 and 544882710). T.B. gratefully acknowledges funding by the DFG (project number: 5041 140321) and is thankful for funding by the LMU Munich's Institutional Strategy LMUexcellent within the framework of the German Excellence Initiative and MS instrumentation by the DFG (INST 86/1800-1 FUGG to Prof. M. Robles. Y.X. acknowledges financial support from the China Scholarship Council (CSC). This work has benefited from a French State grant (Saclay Plant Sciences, reference n° ANR-17-EURE-0007, EUR SPS-GSR) under a France 2030 program (reference n° ANR-11-IDEX-0003), from the facilities and expertise of the I2BC proteomic platform (Proteomic-Gif, SICaPS) supported by IBiSA, Ile-de-France Region, Plan Cancer, CNRS and Paris-Saclay University, and from COST Action CA20113, supported by COST (European Cooperation in Science and Technology). Further support was provided by the French National Research Agency (ANR) through KatNat (ERA-CAPS, ANR-17-CAPS-0001–01) and CanMore (France-Germany PRCI, ANR-20-CE92-0040), and Acetylplast (France-Germany PRCI, ANR-25-CE20-3915) to C.G. We thank Dr. Marcin Luzarowski from Core Facility for Mass Spectrometry and Proteomics of the Center for Molecular Biology (ZMBH) of Heidelberg University for the technical support.

## Author contributions

X.G. performed the majority of the experiments. M.P. contributed to the microscopic analysis. Y.X. and T.B. performed the quantitative proteomic analysis. J.-B.B., T.M., and C.G. performed the N-terminome mass spectrometry analysis. R.H., M.W., and X.G. designed the study, and X.G. and M.W. wrote the manuscript. T.M., T.B., C.G., R.H., and M.W. acquired funding. All authors reviewed the manuscript.

## Funding

## Competing interests

The authors declare no competing interests.
