## [Transparent Peer Review file · Nature Communications]

The ribosome-associated N-terminal acetyltransferase B coordinates global proteostasis and autophagy in plants by creating Ac/N-degrons

Corresponding Author: Dr Markus Wirtz

Version 0:

Reviewer comments:

Reviewer #1

(Remarks to the Author)

This very interesting paper uses a range of different approaches to explore the role(s) of N-terminal acetylation (NTA) catalysed by the NatB complex on the plant proteome. Originally, NTA was thought to stabilize proteins but there is a growing body of evidence that NTA, in certain contexts, acts as a signal for degradation.

Using a CRISPR-generated knockout line (*naa20-cr1*), the authors demonstrated a modest decrease in acetylation of NatB substrates, but some substrates retained NTA status, indicating an unexplained compensation mechanism, which was considered not to involve other known N-acetyl transferases, since they could not rescue the *naa20-cr1* mutant. Polyubiquitination levels were lower in the mutant, suggesting that NatB destabilizes the plant proteome, but the authors did not find strong evidence that NatB regulates the proteome by modulating the stability of all NatB substrates. Indeed, of six NatB substrates tested, only NBR1 (a selective autophagy receptor) was stabilized in *naa20-cr1*.

Importantly, the authors demonstrated that loss of NatB function in the *naa20-cr1* line results in decreased ubiquitin proteasome activity and decreased protein translation. They then explored whether lowered proteasome activity could be caused by enhanced autophagy and provided evidence that NatB-deficient plants display higher autophagic flux, indicating that NatB represses autophagy. This is contrary to the situation in yeast, where NatB is important for autophagy induction. They discovered that NatB-mediated NTA specifically destabilises KIN11, a subunit of the energy sensor kinase, SnRK1. Intriguingly, the closely related subunit, KIN10 was not regulated in the same way. Taken together, the data provide evidence for NatB as a central regulator of UPS-autophagy interplay. Once thought of as discrete processes, it has become increasingly evident that the autophagy and the UPS are linked but the underlying mechanisms have largely remained obscure. Therefore, this study represents an important advance in proteostasis.

Issues to address:

The fact that other NATs don't rescue *naa20-cr1* does not necessarily mean that they don't contribute to the residual NTA activity in the mutant. Perhaps they provide the residual activity but are not active towards the substrates that lack acetylation in *naa20-cr1*? Potentially this could be tested using double mutants although the combinations could be deleterious. At least this possibility could be discussed.

Although NatB does not appear to control stability of RPN12a and V-ATPase (Fig. 3F), they do have reduced NTA according to Table S2. Perhaps this could be mentioned?

AGI codes are provided in Table S3; please could the annotations be added?

Line 202: please clarify whether the six NatB substrates are predicted substrates or whether they have been validated previously.

The manuscript contains several quantified western blots. Whilst several use 3-4 technical replicates, it is not clear whether these experiments have been repeated. Other experiments seem to have little replication: e.g. in Fig. 4A the K63 blot

presented only has one lane per sample; similarly, in Fig. 5G and H, only two replicates are presented, and the loading control looks quite uneven. Have these experiments been repeated? Overall, it was not clear to me how many independent quantification experiments have been performed.

Altered translation is presented at the beginning of the manuscript but is not touched upon in the Discussion. It would be helpful to revisit this. Also, it is mentioned that the potential effect of NatB-regulation of KIN11 activity on TOR kinase remains to be investigated but it would be interesting and relatively straightforward to do a preliminary test with phosphorylation of ribosomal protein S6 (RPS6) as a readout (western with anti-P-RPS6 antibody).

Please check the labelling on Fig. S6F

Methods: please provide dilutions for all antibodies.

Reviewer #2

(Remarks to the Author)

This study by Gong et al is focussed on assessing the role of NATB mediated N-terminal acetylation (NTA) on protein stability and overall proteostasis (via the UPS and autophagy) in Arabidopsis. NTA is a widespread modification that has received increased attention in recent years as more and more functions have been ascribed, many by the authors of this study. The work here shows that NATB influences global protein turnover through a combination of the creation of so-called Ac/N-degrons in certain target proteins (e.g. NBR1 and KIN11, where the data are convincing) and through enhancing autophagy. This is in keeping with some prior reports in yeast and animals connecting NTA to protein destabilisation, but contrasts with findings for NATB substrates in drosophila, and indeed with the effects of NTA-mediated NTA on protein fate in plants. As such the study adds significantly to our knowledge of the intriguing and varied effects of NTA on protein stability, and will be of broad interest to the community. It also provides interesting new insights into the cross talk between UPS and autophagy pathways, and how NTA-mediated control of Kin11 stability, part of the SnRK1 complex, may orchestrate this.

The paper is well written and overall the data/figures are presented in an accessible manner. I have a few comments and queries, listed here in order of appearance in the manuscript.

Line 42: consider changing the word “stunning”.

Line 64: I think the word vital is incorrect here, as this suggests that they are needed, when in fact Arabidopsis is still able to live without them.

For completeness, in the introduction when introducing plant-specific NATB studies, the authors should also cite another of their papers that has focussed on assessing roles for NATB in plants (with respect to reductive stress sensitivity) – Huber et al 2022, *Frontiers in Plant Science*.

Line 82: Here it might be interesting to comment on the fact that the UBR1 protein is not directly conserved in plants (Arabidopsis), where instead the degradation of proteins bearing type 1 and type2 N-degrons is undertaken by separate, plant specific N-recognins (PRT6 and PRT1, respectively). This might also contribute to kingdom specific differences in the effect of NATB on substrate fate.

Line 143: The meaning of “>20% decrease” could be made clearer here. Do you mean that 22 proteins had a more than 20% reduction, or that they had less than 20% levels of acetylation. It was hard to directly match this statement to any of the figures by eye (e.g., 1F and 1G), so it might be better to spell it out more clearly. Related to this, obviously 22 proteins is a very small amount to base proteome-wide conclusions on, so some extra details of what percentage of the total number of potential NATB substrates this covers (i.e., 20% of the whole proteome) would be useful.

Line 157: Please provide some further in-text details of what was actually measured in terms of proteasome stability? Did you measure single or combined protease activities etc. It would be useful to be clearer about this in the main text.

Line 159: Could the authors comment on why the reduced levels of polyubiquitylated proteins are only apparent upon proteasome inhibition (Fig 2B), when the relative proteasome activity (Fig 2A) is significantly reduced even under non-treated conditions? Also, is this result not paradoxical? If natb mutants have reduced proteasome activity, would one not expect to see enhanced polyubiquitylated proteins compared to WT, due to reduced capacity to degrade them? I realise that later in the paper the authors also examine autophagy, and show elevated K63-linked substrates in natb mutants, but it would be good to add some extra information here.

Line 171: should that say “more than twice the percentage of stabilised NATA substrates”?

Line 172: It is interesting that “translation” associated GO terms are up, given that basal translation rates are reduced in the natb. Is this likely linked to an unsuccessful attempt to compensate? Please add some more thoughts/clarification.

Figure 4A: It would be important to repeat this K63 blot across multiple biological replicates, or (ideally) to show it in different natb mutants alleles (CRISPR and T-DNA), as was done for the K48/proteasome assays in Fig. 3.

Figures 4B-D: In the text it says the ATG proteins are significantly enhanced in the mutant – were statistics performed on multiple repeats? It might be better to perform a dilution (e.g., 1x, 0.5x and 0.25x) in both the mutants vs WT to see if any differences are clearer for ATG8, ATG3 and ATG5. By eye think the proposed increase looks to only be subtle, so alternatively I would consider rewording. Did you also look at expression levels by qPCR to see if their increase linked is transcriptional or post-translational changes?

Line 355: This refers to a figure 7E, which is non-existent.

Conclusion: This paper is quite complex with lots of molecular details, and sometimes seemingly contradictory findings. As such, I appreciated the conclusion paragraph, which helped to reiterate and synthesise all the key findings.

Reviewer #3

(Remarks to the Author)

This manuscript reports that loss of NATB activity in *naa20* mutant reduces proteasomal function, resulting in slower degradation of a subset of proteins in Arabidopsis. The authors further show that autophagy activity is elevated in the *naa20* mutant and that NBR1 accumulates due to reduced degradation, although the authors conclude that NBR1 accumulation is not the cause of enhanced autophagy. They also report increased phosphorylation of KIN10 and KIN11 in the *naa20* mutant, along with KIN11 protein accumulation, which leads to elevated autophagy and enhanced tolerance to carbon starvation (dark stress). Although the topic is of some interest, the overall novelty is limited, that prevents the manuscript from being suitable for publication in Nature Communications.

Concerns

1. Limited novelty due to a similar regulatory pathway well established in mammals. In mammalian cells, NAA20 was previously shown to regulate N-terminal acetylation of LKB1, which is the upstream kinase of AMPK (SnRK1 in plants), and thereby inhibiting activation of the LKB1–AMPK pathway (Experimental & Molecular Medicine, 2020, 1831–1844). Thus, NATB functions as a negative regulator of AMPK signaling. In this study, the authors report a similar mechanism in plants, proposing that NAA20 regulates KIN11 stability via N-terminal acetylation and thereby negatively regulates SnRK1 signaling. Because this regulatory logic has already been established in mammals, the conceptual novelty of this work is limited. Moreover, in plants, NATB may regulate SnRK1 signaling through SnAK2, the upstream kinase of SnRK1. The authors should experimentally test whether NATB affects SnAK2 activity.
2. In the *naa20* mutant, only ~30% of the accumulated proteins are predicted NATB substrates, whereas ~60% are predicted NATA substrates (Fig. 2). Likewise, among proteins with reduced ubiquitination in *naa20*, ~60% are NATA substrates (Fig. 3B). How do the authors explain that depletion of NATB more substantially affect NATA substrates? Given that most affected proteins are not NATB substrates, performing GO enrichment analysis using the entire set of proteins (Fig. 3C–E) may not be appropriate. I recommend performing GO analysis specifically on the predicted NATB substrates.
3. The authors conclude that NBR1 accumulates in *naa20* due to impaired proteasomal degradation, and that increased autophagy does not counteract this accumulation. However, this cannot be firmly concluded without directly testing NBR1 turnover when proteasome activity is inhibited (e.g., MG132 treatment) or when autophagy is blocked (e.g., ConA treatment). The authors should analyze NBR1 degradation dynamics in *naa20* under both/ and either MG132 and ConA treatments.
4. N-terminal acetylation can influence not only protein stability but also subcellular localization and protein–protein interactions. Does loss of N-terminal acetylation alter NBR1 localization or its interaction with ATG8?
5. The confocal images in Fig. 4E include excessively large fields of view, making autophagosomes difficult to identify. Higher-magnification images are needed. Additionally, in the WT Dark + ConA (6 h) panel, two central columns of cells show no autophagosomes, which contradicts previously published studies. Moreover, the authors should perform a rigorous GFP-cleavage assay to accurately quantify autophagic flux during a time-course of carbon starvation treatment.
6. Both KIN10 and KIN11 are predicted NATB substrates, and both show increased phosphorylation in the *naa20* mutant. However, only KIN11 protein accumulates. What accounts for this striking difference? How do variations in the N-terminal sequences of KIN10 and KIN11 lead to such divergent outcomes? Additionally, how does loss of N-terminal acetylation influence phosphorylation levels of these kinases?
7. Are the expression levels of known KIN11 target genes increased in *naa20*? This would demonstrate functional activation of KIN11.
8. Is KIN11 a direct NATB substrate? Is it possible to perform in vitro acetylation assays to directly test whether KIN11 is acetylated by NATB?
9. Do KIN10 and KIN11 undergo ubiquitin-mediated degradation? If so, how does KIN11 N-terminal acetylation affect its ubiquitination and turnover?
10. Multiple studies have shown that KIN10 is the major catalytic subunit of SnRK1 protein kinase complex, and *kin10* mutants are hypersensitive to abiotic stresses. Why does *kin10* not display stress-sensitive phenotypes in this study?
11. What are the weak protein bands observed in the WT sample in Fig. S1B?
12. Because proteasome activity is reduced in *naa20*, one would expect accumulation of ubiquitinated proteins under normal conditions. Why is this not observed?
13. Does NBR1 accumulate in the proteomics dataset of the *naa20* mutant? This should be reported explicitly.
14. Figures 4A–D show levels of autophagy-related proteins, but the treatment conditions are not specified. These must be clearly stated in the figure legends or main text.
15. Given the increased autophagy activity in *naa20*, do these plants exhibit delayed senescence, particularly under short-day conditions?
16. The statement “only three point 7 (3.788) proteins were identified” should be a typographical error.

Version 1:

Reviewer comments:

Reviewer #1

(Remarks to the Author)

I am satisfied that the authors have addressed my comments. They have also provided several pieces of additional data to address the comments of the other reviewers.

Reviewer #2

(Remarks to the Author)

I am satisfied with the revised version of the manuscript submitted to Nature Communications. The authors have adequately addressed all of my queries and comments.

Reviewer #3

(Remarks to the Author)

The authors have made an effort to address the previous concerns. However, several critical issues remain regarding the interpretation of the autophagy data and the genetic evidence supporting the KIN11 pathway. The following points should be addressed to improve the quality of the manuscript:

1. The authors conclude that autophagy is upregulated in the *naa20* mutant based on increased ATG8 lipidation, ATG5-ATG12 conjugation, and the presence of autophagic bodies. However, these markers can be misinterpreted if autophagic flux (the complete degradation process) is not properly distinguished from the mere accumulation of autophagosomes.

(1) ATG8 Lipidation: Increased levels of ATG8-PE are often observed in mutants where autophagosome-vacuole fusion is disrupted (reduced flux), not just when autophagy is induced. Furthermore, the protein bands in Figure 4c are ambiguous; the lower faint band may not represent the lipidated form. The authors must include an *atg5* or *atg7* mutant (deficient in ATG8-PE formation) as a negative control to definitively identify the ATG8-PE band.

(2) ATG5-ATG12 Conjugation: In plant systems, it is established that most endogenous ATG5 exists in the conjugated (ATG5-ATG12) form (see PMC3967041). In Figure 4e, the majority of ATG5 appears as a free protein. This contradicts established literature and requires a technical explanation or re-validation.

(3) Confocal Imaging and Vacuolar Localization: The "dots" identified as autophagic bodies in Figure 4g appear to be localized in the cytoplasm rather than within the vacuole. If these structures are cytoplasmic, it suggests a block in trafficking/fusion, implying decreased rather than increased flux. The authors should redo confocal analysis using root elongation cells treated with Concanamycin A to stabilize autophagic bodies within the vacuole, and take images to demonstrate the vacuolar autophagic bodies following the procedures shown in some previous publications (e.g., Zhao et al., J Cell Biol, 2022; Stephani et al., eLife, 2020). Moreover, to provide definitive proof of autophagic flux, a standard GFP-cleavage immunoblotting assay should be performed.

2. The authors claim that the *kin11* single mutant is sensitive to dark treatment and posits that the dark-tolerant phenotype of *naa20* is due to elevated KIN11 (but not KIN10) activity. To strengthen this causal link, the authors should provide evidence from gain-of-function lines. Have the authors tested the dark-tolerance phenotypes of KIN11-OE or KIN11(D2P)-OE (phosphomimetic) overexpression plants?

Version 2:

Reviewer comments:

Reviewer #3

(Remarks to the Author)

The authors have made efforts to address my concerns by including new experimental data. I have no more questions and would like to suggest acceptance of this interesting story.

REVIEWER COMMENTS

Reviewer #1 (Remarks to the Author):

This very interesting paper uses a range of different approaches to explore the role(s) of N-terminal acetylation (NTA) catalysed by the NatB complex on the plant proteome. Originally, NTA was thought to stabilize proteins but there is a growing body of evidence that NTA, in certain contexts, acts as a signal for degradation.

Using a CRISPR-generated knockout line (*naa20-cr1*), the authors demonstrated a modest decrease in acetylation of NatB substrates, but some substrates retained NTA status, indicating an unexplained compensation mechanism, which was considered not to involve other known N-acetyl transferases, since they could not rescue the *naa20-cr1* mutant. Polyubiquitination levels were lower in the mutant, suggesting that NatB destabilizes the plant proteome, but the authors did not find strong evidence that NatB regulates the proteome by modulating the stability of all NatB substrates. Indeed, of six NatB substrates tested, only NBR1 (a selective autophagy receptor) was stabilized in *naa20-cr1*.

Importantly, the authors demonstrated that loss of NatB function in the *naa20-cr1* line results in decreased ubiquitin proteasome activity and decreased protein translation. They then explored whether lowered proteasome activity could be caused by enhanced autophagy and provided evidence that NatB-deficient plants display higher autophagic flux, indicating that NatB represses autophagy. This is contrary to the situation in yeast, where NatB is important for autophagy induction. They discovered that NatB-mediated NTA specifically destabilises KIN11, a subunit of the energy sensor kinase, SnRK1. Intriguingly, the closely related subunit, KIN10 was not regulated in the same way. Taken together, the data provide evidence for NatB as a central regulator of UPS-autophagy interplay. Once thought of as discrete processes, it has become increasingly evident that the autophagy and the UPS are linked but the underlying mechanisms have largely remained obscure. Therefore, this study represents an important advance in proteostasis.

Issues to address:

The fact that other NATs don't rescue *naa20-cr1* does not necessarily mean that they don't contribute to the residual NTA activity in the mutant. Perhaps they provide the residual activity but are not active towards the substrates that lack acetylation in *naa20-cr1*? Potentially this could be tested using double mutants although the combinations could be deleterious. At least this possibility could be discussed.

Response: We agree with the reviewer that this is a possible scenario, which we have discussed in a new paragraph in the Discussion section (Lines 415-432).

Although NatB does not appear to control stability of RPN12a and V-ATPase (Fig. 3F), they do have reduced NTA according to Table S2. Perhaps this could be mentioned?

Response: As suggested by reviewer 1, we have rephrased the corresponding statements in the improved manuscript version and highlighted this relevant information and its consequence for the regulation of protein stability by N-terminal acetylation of NatB substrates (Lines 207-209).

AGI codes are provided in Table S3; please could the annotations be added?

Response: We thank the reviewer for carefully reading our manuscript and for this helpful suggestion. The annotations have been added to Supplementary Table 3 and Supplementary Table 4.

Line 202: please clarify whether the six NatB substrates are predicted substrates or whether they have been validated previously.

Response: Among these six NatB substrates, V-ATPase and RPN12a have been experimentally confirmed in our study using a proteomics approach (Supplementary Table 2), whereas the other four are predicted NatB substrates based on their sequences. We have rephrased the corresponding sentences and added the information in Lines 205-209. The identity of NBR1 as a NatB substrate was later experimentally confirmed by in vitro enzymatic assays using immunoprecipitated plant NatB (Supplementary Fig. 4b, Lines 216-221).

The manuscript contains several quantified western blots. Whilst several use 3-4 technical replicates, it is not clear whether these experiments have been repeated. Other experiments seem to have little replication: e.g. in Fig. 4A the K63 blot presented only has one lane per sample; similarly, in Fig. 5G and H, only two replicates are presented, and the loading control looks quite uneven. Have these experiments been repeated? Overall, it was not clear to me how many independent quantification experiments have been performed.

Response: All experiments were repeated at least three times. In most cases, each independent repetition included multiple replicates from different individuals except for some IP experiments (e.g., Fig. 4a) or when too many genotypes needed to be investigated (e.g., Fig. 5g, h). We have specified the number of replicates throughout the figure legends and added the information in the “Source Data File”. For Fig. 4A, we have prepared a new graph showing the quantification of three independent repetitions with one replicate and added the calculated p-value. The immunoblots of repetition 2 and 3 are included in the Source Data file. The blot shown in the manuscript represents the blot that is closest to the calculated mean value for the difference between K63-polyubiquitination in *naa20* mutants and the wild type.

Altered translation is presented at the beginning of the manuscript but is not touched upon in the Discussion. It would be helpful to revisit this. Also, it is mentioned that the potential effect of NatB-regulation of KIN11 activity on TOR kinase remains to be investigated but it would be interesting and relatively straightforward to do a preliminary test with phosphorylation of ribosomal protein S6 (RPS6) as a readout (western with anti-P-RPS6 antibody).

Response: We appreciate the reviewers' comments and suggestions. As suggested, we detected TOR activity in the *natb* mutants by monitoring the ratio of p-RPS6 to RPS6 levels as a readout (New Supplementary Fig. 6e). The results revealed reduced p-RPS6 levels in the *natb* mutants compared to the wild type, suggesting a lower TOR activity, which might help explain the generally lowered translation rate in the *natb* mutants (Lines 332-335). These novel findings are discussed in a new paragraph of the Discussion section, highlighting the NatB depletion-induced changes in translation (Lines 492-495).

Please check the labelling on Fig. S6F

Response: We thank the reviewer for carefully reading the manuscript. The mistake in the description of the x-axis has been corrected (New Supplementary Fig. 6h).

Methods: please provide dilutions for all antibodies.

Response: This information has been added to the improved version of the manuscript (Lines 603-616).

Reviewer #2 (Remarks to the Author):

This study by Gong et al is focussed on assessing the role of NATB mediated N-terminal acetylation (NTA) on protein stability and overall proteostasis (via the UPS and autophagy) in Arabidopsis. NTA is a widespread modification that has received increased attention in recent years as more and more functions have been ascribed, many by the authors of this study. The work here shows that NATB influences global protein turnover through a combination of the creation of so-called Ac/N-degrons in certain target proteins (e.g. NBR1 and KIN11, where the data are convincing) and through enhancing autophagy. This is in keeping with some prior reports in yeast and animals connecting NTA to protein destabilisation, but contrasts with findings for NATB substrates in Drosophila, and indeed with the effects of NTA-mediated NTA on protein fate in plants. As such the study adds significantly to our knowledge of the intriguing and varied effects of NTA on protein stability, and will be of broad interest to the community. It also provides interesting new insights into the cross talk between UPS and autophagy pathways, and how NTA-mediated control of Kin11 stability, part of the SnRK1 complex, may orchestrate this.

The paper is well written and overall the data/figures are presented in an accessible manner. I have a few comments and queries, listed here in order of appearance in the manuscript.

Line 42: consider changing the word “stunning”.

Response: We appreciate the reviewers' suggestion. The word “stunning” has been replaced by “increased” (Line 42).

Line 64: I think the word vital is incorrect here, as this suggests that they are needed, when in fact Arabidopsis is still able to live without them.

Response: The word “vital” has been replaced by “viable” (Line 60).

For completeness, in the introduction when introducing plant-specific NATB studies, the authors should also cite another of their papers that has focussed on assessing roles for NATB in plants (with respect to reductive stress sensitivity) – Huber et al 2022, Frontiers in Plant Science.

Response: We appreciate the reviewers' suggestion. The reference has been added (Line 61).

Line 82: Here it might be interesting to comment on the fact that the UBR1 protein is not directly conserved

in plants (*Arabidopsis*), where instead the degradation of proteins bearing type 1 and type 2 N-degrons is undertaken by separate, plant specific N-recognins (PRT6 and PRT1, respectively). This might also contribute to kingdom specific differences in the effect of NATB on substrate fate.

Response: We appreciate the reviewers' suggestion and added the proposed information and references to Lines 77-80.

Line 143: The meaning of “>20% decrease” could be made clearer here. Do you mean that 22 proteins had a more than 20% reduction, or that they had less than 20% levels of acetylation. It was hard to directly match this statement to any of the figures by eye (e.g., 1F and 1G), so it might be better to spell it out more clearly. Related to this, obviously 22 proteins is a very small amount to base proteome-wide conclusions on, so some extra details of what percentage of the total number of potential NATB substrates this covers (i.e., 20% of the whole proteome) would be useful.

Response: We thank the reviewer for pointing this out. We meant that the NTA level of these proteins was more than 20% lower in *naa20* mutants than in the wild type. We have rephrased the corresponding sentences to make this statement more straightforward (Lines 137-142). In fact, we have identified only the N-termini of 33 NatB substrates, whose NTA levels are more than 20% lower in *naa20* mutants, out of the potential 8948 NatB substrates of the whole proteome. This discrepancy arises because we need to identify and quantify the acetylated and the non-acetylated N-terminal peptides for a given protein in one of the replicates for each genotype to determine the N-terminal acetylation level of the respective proteins. Indeed, the number of quantified protein N-termini is low, but the method used here is the current state-of-the-art.

Line 157: Please provide some further in-text details of what was actually measured in terms of proteasome stability? Did you measure single or combined protease activities etc. It would be useful to be clearer about this in the main text.

Response: We apologize that this information was hidden in the material and method section. We have determined the chymotrypsin-like protease activity of the proteasome as a proxy for the overall proteasome abundance. In this assay, the extractable proteasome activity in the soluble protein fraction is determined by monitoring the release of fluorescent AMC after the proteasome cleaves the model substrate Z-Leu-Leu-Leu-AMC. We have rephrased the corresponding sentence and added the relevant information in the result text (Lines 154-157).

Line 159: Could the authors comment on why the reduced levels of polyubiquitylated proteins are only apparent upon proteasome inhibition (Fig 2B), when the relative proteasome activity (Fig 2A) is significantly reduced even under non-treated conditions?...

Response: This is because, in addition to the relative proteasome activity, the endogenous polyubiquitination rate is decreased in *natb* mutants. As a consequence, the steady-state polyubiquitin level of the proteome in the WT and *natb* is comparable. Please note that proteasome inhibition also results in an increase of poly-ubiquitinated proteins in *natb* mutants, but this increase is lower than in the wild type. Thus, *natb* mutants possess a lower endogenous polyubiquitination rate. One of our claims is that this lowered endogenous polyubiquitination rate is caused by reduced activity of N-recognins in the

Ac/N-degron pathway, as these N-recognins do not recognize the non-acetylated NatB substrates in *natb* mutants.

...Also, is this result not paradoxical? If *natb* mutants have reduced proteasome activity, would one not expect to see enhanced polyubiquitylated proteins compared to WT, due to reduced capacity to degrade them? I realise that later in the paper the authors also examine autophagy, and show elevated K63-linked substrates in *natb* mutants, but it would be good to add some extra information here.

Response: This assumption would be correct if the translation rate and the E3-ligase activity (defining the endogenous polyubiquitination rate) in *natb* mutants were not affected. We show in Fig. 2a that *natb* mutants exhibit lower extractable proteasome activity, and at the same time, a lower global translation rate (Fig. 2d). Since global degradation and global translation in the *natb* mutant are decreased, the steady-state proteome level (Fig. 2e) is almost unaffected, and the “global amount” of ubiquitinated proteins is not affected (Fig. 2c, compare WT DMSO with *naa20* DMSO). Furthermore, the accumulation of ubiquitinated proteins also depends on the endogenous E3 ligase activity. We claim that lowered NTA of NatB substrates causes less ubiquitination of NatB substrates. Even if the proteasome activity is decreased, polyubiquitinated proteins should only accumulate in *naa20*, if these proteins are translated and ubiquitinated with the same rate. Both translation (Fig. 2d) and global endogenous polyubiquitination rate (Fig. 2c, now compare the difference between DMSO and MG132 in the wild type versus the difference between DMSO and MG132 in *naa20*) are decreased in *naa20*. As a consequence, we do not observe a significant accumulation of polyubiquitinated proteins in *naa20*.

Line 171: should that say “more than twice the percentage of stabilised NATA substrates”?

Response: We apologize for failing to explain this comparison in the previous version of the manuscript. In the wild type, only 16 % of stabilized proteins were NatB substrates. In the *natb* mutant, 30 % of stabilized proteins are NatB substrates. This is almost twice the percentage of NatB proteins stabilized in the wild type, and strongly suggests that, in general, NatB substrates tend to be more stable in *natb* mutants. The improved version of the manuscripts provides more information, hopefully making this comparison more straightforward to understand (Lines 169-174).

Line172: It is interesting that “translation” associated GO terms are up, given that basal translation rates are reduced in the *natb*. Is this likely linked to an unsuccessful attempt to compensate? Please add some more thoughts/clarification.

Response: We greatly appreciate the reviewers’ helpful comments to improve the manuscript’s readability and have added this possibility to the results section (Lines 178-179).

Figure 4A: It would be important to repeat this K63 blot across multiple biological replicates, or (ideally) to show it in different *natb* mutants alleles(CRISPR and T-DNA), as was done for the K48/proteasome assays in Fig. 3.

Response: We fully agree with the reviewer about the importance of independent repetitions. The improved Figure 4A now shows the results from three independent replicates. A representative Western blot is shown in addition to allow the reader to evaluate this assay. The complete blots for the other replicates have been shown in the manuscript's Source Data file.

Figures 4B-D: In the text it says the ATG proteins are significantly enhanced in the mutant – were statistics performed on multiple repeats? It might be better to perform a dilution (e.g., 1x, 0.5x and 0.25x) in both the mutants vs WT to see if any differences are clearer for ATG8, ATG3 and ATG5. By eye think the proposed increase looks to only be subtle, so alternatively I would consider rewording....

Response: We apologize for the potentially unclear statement in the previous manuscript version. Yes, the experiments have been independently repeated and always showed an increase of ATG-related proteins in the *natb* mutants. We have specified the number of replicates in the figure legends and added the p-values for the exemplary shown repetitions, which included four independent biological replicates, to the “Source Data File”. For ATG8, we quantified the lipidated ATG8 (ATG-PE), a post-translationally lipidated proteoform of ATG8 covalently conjugated to phosphatidylethanolamine. This ATG8 proteoform is essential for autophagosome membrane expansion. In case of ATG5, we quantified the amount of ATG5 bound in the ATG5-ATG12 conjugated complex amount, which is essential for autophagosome formation by acting as an E3-like enzyme that promotes ATG8 lipidation.

....Did you also look at expression levels by qPCR to see if their increase linked is transcriptional or post-translational changes?

Response: As suggested, we performed qPCR on the autophagy-related genes tested here. We did not find a significant increase in the steady-state transcript levels of these genes in *naa20cr1*, suggesting that the rise in ATG proteins is caused by post-transcriptional regulation or post-translational modifications. The novel results are shown in Figure 4F and described in the results text (Lines 272-274).

Line 355: This refers to a figure 7E, which is non-existent.

Response: We thank the reviewer for carefully reading our manuscript and pointing out this mistake. Indeed, we wanted to refer to Supplementary Fig.7c and Supplementary Fig.7d. This mistake is corrected in the improved manuscript version.

Conclusion: This paper is quite complex with lots of molecular details, and sometimes seemingly contradictory findings. As such, I appreciated the conclusion paragraph, which helped to reiterate and synthesise all the key findings.

Response: We thank the reviewer for appreciating our efforts to improve the readability of this complex manuscript.

Reviewer #3 (Remarks to the Author):

This manuscript reports that loss of NATB activity in *naa20* mutant reduces proteasomal function, resulting in slower degradation of a subset of proteins in Arabidopsis. The authors further show that autophagy activity is elevated in the *naa20* mutant and that NBR1 accumulates due to reduced degradation, although the authors conclude that NBR1 accumulation is not the cause of enhanced autophagy. They also report increased phosphorylation of KIN10 and KIN11 in the *naa20* mutant, along with KIN11 protein accumulation, which leads to elevated autophagy and enhanced tolerance to carbon

starvation (dark stress). Although the topic is of some interest, the overall novelty is limited, that prevents the manuscript from being suitable for publication in Nature Communications.

Concerns

1. Limited novelty due to a similar regulatory pathway well established in mammals. In mammalian cells, NAA20 was previously shown to regulate N-terminal acetylation of LKB1, which is the upstream kinase of AMPK (SnRK1 in plants), and thereby inhibiting activation of the LKB1–AMPK pathway (Experimental & Molecular Medicine, 2020, 1831–1844). Thus, NATB functions as a negative regulator of AMPK signaling. In this study, the authors report a similar mechanism in plants, proposing that NAA20 regulates KIN11 stability via N-terminal acetylation and thereby negatively regulates SnRK1 signaling. Because this regulatory logic has already been established in mammals, the conceptual novelty of this work is limited.

Response: We thank the reviewer for the opportunity to reaffirm the novelty of our work. We agree with the reviewer that NatB-mediated regulation of energy-sensing kinases has been previously reported in the mammalian system via the LKB1–AMPK–mTOR axis [1]. However, we would like to emphasize that

(1) Although LKB1 was identified as a substrate of NatB in vitro, Jung and coworkers failed to confirm the decrease in LKB1 acetylation level in NatB-depleted mammalian cells. Thus, a direct action of NatB on LKB1 is still questionable. Furthermore, the authors do not even propose a direct action of NatB on the catalytic AMPK subunits [1], as we did. As a consequence, their proposed mechanism for NatB-mediated AMPK regulation is fundamentally different from the regulation we propose for SnRK1 by NatB in plants.

(2) Human AMPK is fundamentally different from plant SnRK1 in its regulation. In contrast to the **AMP**-activated **Protein Kinase** (AMPK), SnRK1 is not activated by AMP at all. Plant SnRK1 senses Trehalose-6-P levels and redox changes [2, 3], which is explained by the substantially different motif composition in the C-terminal located regulatory domain of the catalytically active α -subunits of SnRK1 when compared to AMPK. On top of these differences, the regulatory β subunit of AMPK is replaced by the $\beta\gamma$ -subunit in the plant SnRK1 complex [4]. Thus, one would not expect a conserved mechanism for regulating AMPK/SnRK1 in eukaryotes between the red and green lineages.

(3) In yeast, NatB is essential for autophagy; thus, *natb* yeast mutants are deficient in autophagy [5]. In stark contrast, NatB depletion in humans induces autophagy, potentially via the proposed LKB1-AMPK axis [1].

Given the fundamental differences in AMPK1 and autophagy regulation between fungi, animals, and plants, we do not believe that the study by Jung et al. undermines the novelty of our manuscript. In contrast to Jung et al. and Shen et al., we demonstrate that NatB inhibits autophagy in plants by providing direct evidence that the absence of NTA of KIN11, a subunit of the energy sensor SnRK1, protects it from UPS-mediated Ac/N-degron pathway. Our findings not only identify the role of NTA in regulating autophagy in plants for the first time but also establish NatB as a central regulator of UPS–autophagy interplay.

In addition to this finding, we demonstrate, for the first time, that NatB controls general protein turnover in a eukaryote by imprinting a destabilizing mark on a subset of its substrates. We strongly believe that these findings are of significant interest to the broad readership of Nature Communications. We have added a new paragraph in the Discussion section to summarize our key findings (Lines 486-498).

...Moreover, in plants, NATB may regulate SnRK1 signaling through SnAK2, the upstream kinase of SnRK1. The authors should experimentally test whether NATB affects SnAK2 activity.

Response: The N-terminus of LKB1 is highly conserved in humans and animals. Its sequence suggests that LKB1 is a NatB substrate as it starts with ME [1]. In stark contrast, the only known upstream kinases of SnRK1 in plants, SnRK1 activating kinase 1 and 2 (SnAK1/2), exhibit an N-terminus (MF, see Response Fig. 1) that is not accepted by NatB in plants [6], humans [7], or yeast [8]. Whether NatB affects SnAK1/2 activity through a mechanism other than direct acetylation is an interesting question, but it is far beyond the scope of this study.

Response Fig. 1 Sequence comparison between human LKB1 and the plant SnRK1-activating kinases SnAK1 and SnAK2.

2. In the *naa20* mutant, only ~30% of the accumulated proteins are predicted NATB substrates, whereas ~60% are predicted NATA substrates (Fig. 2). Likewise, among proteins with reduced ubiquitination in *naa20*, ~60% are NATA substrates (Fig. 3B). How do the authors explain that depletion of NATB more substantially affect NATA substrates?

Response: We understand the reviewer's concerns regarding the number of stable NatA substrates in *natb* mutants. In this context, it is important to mention that NatA substrates are often stable and abundant housekeeping proteins (e.g., the cysteine synthesizing protein OAS-TL A [9]). For that reason, they are frequently overrepresented in quantitative proteomics studies. In our quantitative proteomics approach 61% of all identified proteins in the wild type are NatA substrates (Figure 2E). NatA acetylates only 40% of the proteome based on predictions of substrate N-termini encoded in the genome.

The reviewer is correct that the number of affected NatA substrates in *natb* mutants is high. However, the percentage of stable NatA substrates is substantially lower in *natb* mutants (52%) than in the wild type (68%). This decrease of NatA substrates is caused by the higher number of stable NatB substrates in *natb* mutants. The number of non-NatB substrate proteins is still high because NatA substrates are, in general, stable proteins in the wild type. Moreover, the Ubi capture followed by mass spectrometry analysis clearly demonstrates that NatB destabilizes the plant proteome. In this approach, we found that out of 314 significantly changed polyubiquitinated proteins, 310 (99%) were more ubiquitinated in the wild type than in *naa20-cr1* (Fig. 3a and Supplementary Table 4). These results suggest that NatB may not only regulate the plant proteome by modulating the stability of individual NatB substrates, but also indirectly by modulating the UPS. In line with this hypothesis, 42% of the identified proteins in the enriched GO term “ubiquitin-dependent protein catabolic process” are NatB substrates (Fig. 3e). This finding suggests that NatB substrates play a critical role in controlling protein homeostasis. To explain this apparent discrepancy, we have added a more comprehensive description and discussion of these findings in the results section (Lines 200-204, 456-461).

Given that most affected proteins are not NATB substrates, performing GO enrichment analysis using the entire set of proteins (Fig. 3C–E) may not be appropriate. I recommend performing GO analysis specifically on the predicted NATB substrates.

Response: We thank the reviewer for this suggestion. In fact, we thought along the same lines. We performed this GO analysis on the predicted NatB substrates right at the beginning, but were hampered by the technical difficulty that the number of NatB substrates (49) was too low to yield meaningful Gene Enrichment Ontology results. The algorithm needs a certain number of proteins to find pathways in which these proteins are enriched. Therefore, we performed GO analysis on all identified proteins because this approach allowed us to address how NatB affects the composition of the overall plant proteome.

3. The authors conclude that NBR1 accumulates in *naa20* due to impaired proteasomal degradation, and that increased autophagy does not counteract this accumulation. However, this cannot be firmly concluded without directly testing NBR1 turnover when proteasome activity is inhibited (e.g., MG132 treatment) or when autophagy is blocked (e.g., ConA treatment). The authors should analyze NBR1 degradation dynamics in *naa20* under both/ and either MG132 and ConA treatments.

Response: We thank the reviewers for this interesting suggestion for strengthening our claim. As suggested, we analyzed NBR1 degradation dynamics in *naa20-cr1* under both/ and either MG132 and ConA treatments. When protein synthesis was inhibited by Cycloheximide (CHX) for 6 hours, the steady-state level of NBR1 dropped to 45% of the control level. Both MG132 (78%) and ConA (60%) alone partially inhibit NBR1 degradation, with MG132 exhibiting a more pronounced effect. When MG132 and ConA were applied together, NBR1 degradation was inhibited entirely (106%, Supplementary Fig. 4e). These findings indicate that NBR1 is degraded via both the UPS and autophagy pathways *in vivo*. Under the non-stressed conditions applied here, in which autophagy is not required for nutrient recycling, NBR1 turnover was slightly more dependent on the UPS. We report these findings in the result text (Lines 233-240) and Supplementary Fig. 4e.

4. N-terminal acetylation can influence not only protein stability but also subcellular localization and protein–protein interactions. Does loss of N-terminal acetylation alter NBR1 localization or its interaction with ATG8?

Response: We agree with the reviewer that N-terminal acetylation can influence not only protein stability but also subcellular localization and protein–protein interactions. In the revised manuscript, we investigated NBR1 localization using stable transgenic *NBR1-mCherry-GFP/WT* and *NBR1-mCherry-GFP/naa20-cr1* lines. In wild-type root tip cells, NBR1 exhibits a cytosolic, punctate localization pattern consistent with previous studies [10]. In the *naa20-cr1* mutant, the localization pattern of NBR1 remains unchanged, indicating that the absence of NTA does not affect NBR1 localization (new Supplementary Fig. 4g).

To investigate whether loss of N-terminal acetylation alters the interaction between NBR1 and ATG8, we performed a Co-IP assay using stable *NBR1-HA/WT* and *NBR1-HA/naa20-cr1* transgenic plants. NBR1 still interacted with ATG8 in *naa20-cr1*, suggesting that the absence of NTA does not affect the interaction between NBR1 and ATG8 (new Supplementary Fig. 4h). This is consistent with the established concept that

NBR1 interacts with ATG8 via its AIM (ATG8-interacting motif) domain located in the C-terminal region of NBR1. These novel results are included in Supplementary Fig. 4g and Supplementary Fig. 4h and reported in (Lines 245-255).

5. The confocal images in Fig. 4E include excessively large fields of view, making autophagosomes difficult to identify. Higher-magnification images are needed. Additionally, in the WT Dark + ConA (6 h) panel, two central columns of cells show no autophagosomes, which contradicts previously published studies. Moreover, the authors should perform a rigorous GFP-cleavage assay to accurately quantify autophagic flux during a time-course of carbon starvation treatment.

Response: We apologize for the excessively large fields of view images and the misleading image of the WT Dark+ConA in the previous manuscript version. We included higher-magnification images in the improved manuscript version, replacing the previous WT Dark+ConA image. The results presented in these higher-quality images are consistent with previous studies.

Regarding the assessment of elevated autophagy, we have included **six** independent assays to quantify autophagy-induction in *naa20*: (1) Phenotypic analysis of nutrient-starved or elongated darkness-treated WT, *naa20*, and *naa20atg* double mutants (Figure 5); (2) Confocal laser scanning microscopy-based analyses of the GFP-ATG8a marker in WT, *naa20*, and *naa20atg5* backgrounds (Figure 4G); (3) Immunological detection of ATG3 protein levels in WT and *natb* mutants (Figure 4D); (4) Demonstration of enhanced ATG8 lipidation in *natb* mutants (Figure 4C); (5) Quantification of ATG5-ATG12 complex formation in WT and *natb* mutants (Figure 4E); (6) In addition, we showed increased K63-polyubiquitination levels in *naa20-cr1* (Figure 4A). Therefore, we believe that the conclusion that autophagy is induced in *natb* mutants is supported by a sufficient number of independent, accepted assays outlined in the “Guidelines for the use and interpretation of assays for monitoring autophagy (4th edition)” and the review method paper “Studying plant autophagy: challenges and recommended methodologies” [11, 12].

6. Both KIN10 and KIN11 are predicted NATB substrates, and both show increased phosphorylation in the *naa20* mutant. However, only KIN11 protein accumulates. What accounts for this striking difference?...

Response: We thank the reviewer for the opportunity to elaborate on this indeed striking difference. The team of authors was also surprised by this difference before we carefully evaluated the primary sequence of KIN10 and KIN11. Both proteins are almost identical, with only a few sequence differences, clustered in three domains. One of these domains is the N-terminus (See Supplementary Fig. 6f and below).

Response Fig. 2 Sequence comparison between two catalytic subunits of the plant SnRK1 complex, KIN10 and KIN11.

The substantially different amino acid sequence (Residues 3 to 11) in this domain most likely evolved to enable the differential degradation of both isoforms by the Ac/N-degron pathway, in which specific E3-ubiquitin ligases (N-recognins) recognize specifically the N-terminus. N-recognins have been shown to distinguish between non-acetylated and acetylated N-termini and are highly selective for the amino acid sequences at the N-terminus (reviewed in [13]). Consequently, the substantial changes at the N-terminus of both KIN isoforms will cause recognition by different N-recognins, or only recognition of KIN11 but not KIN10 by a so far unidentified N-recognin. An example in which KIN11 is specifically degraded while KIN10 remains stable is the phosphate starvation response [14].

...How do variations in the N-terminal sequences of KIN10 and KIN11 lead to such divergent outcomes?

Response: How variations in the N-terminal KIN10/11 sequence can interfere with the selective binding to the N-recognin depends on structural features of the N-terminus and the N-recognin [15], which we cannot predict without knowing the responsible N-recognin. However, examples of specific binding motifs that depend on the addition of only 42 Da (acetyl group) at the N-terminus are widespread [16-19, reviewed in 20]. It is thus evident that significant changes in the surrounding amino acid sequence will interfere with the selective binding of a substrate to its specific N-recognin [reviewed in 13, 15].

...Additionally, how does loss of N-terminal acetylation influence phosphorylation levels of these kinases?

Response: This is an interesting question that we will address in a separate study with a new PhD candidate, as it is outside the scope of this study. This comprehensive study aims to understand the function of NatB in global protein turnover and demonstrates that the NTA of several NatB substrates destabilizes them via the Ac/N-degron pathway. It furthermore connects the NTA-dependent differences in the stability of the two SnRK1 isoforms to autophagy induction and diverse stress responses.

7. Are the expression levels of known KIN11 target genes increased in *naa20*? This would demonstrate functional activation of KIN11.

Response: As suggested, we examined the expression levels of *DIN1*, *DIN6*, *DIN10*, and *bZIP63*, a transcription factor phosphorylated by SnRK1 that regulates transcription of the aforementioned *DIN* genes. The expression levels of *DIN1*, *DIN6*, and *DIN10* were significantly elevated in the *naa20* mutant, while *bZIP63* expression remained unchanged. These findings suggest that increased SnRK1 activity in *naa20* activates the expression of downstream energy-related genes. The absent transcriptional induction of *bZIP63* suggests that *bZIP63* activation occurs at the post-translational level (e.g., phosphorylation by stabilized KIN11) rather than the transcriptional level. We have included these results in the revised manuscript (Lines 327-332 and Supplementary Fig. 4d).

8. Is KIN11 a direct NATB substrate? Is it possible to perform in vitro acetylation assays to directly test whether KIN11 is acetylated by NATB?

Response: We agree with the reviewer that it is crucial to show that KIN11 is a direct substrate of NatB. For that reason, we showed that immune-precipitated plant NatB acetylates the N-terminus of KIN11 *in vitro* (Fig. 6b). In addition, we quantified the N-terminal acetylation level of KIN11 by quantitative mass spectrometry of immune-precipitated KIN11-HA expressed in the wild type and *naa20-cr1*. We found that N-terminal acetylation of KIN11 is substantially reduced in *naa20-cr1* when compared to wild type (old Figure S6E, F -> now Supplementary Fig. 6g, h). The transcript level of KIN11 was not affected in *naa20-cr1* (old Figure S6G -> now Supplementary Fig. 6i).

9. Do KIN10 and KIN11 undergo ubiquitin-mediated degradation?

Response: In the original manuscript, we showed that KIN11-GFP is polyubiquitinated in the wild type and that the degree of polyubiquitination is significantly lower in *naa20-cr1* (Fig. 6g). Furthermore, we inhibited N-terminal acetylation of KIN11-GFP expressed in the wild type by genetic engineering. The mutagenized KIN11^{D2P} protein was substantially less poly-ubiquitinated in the wild type (Fig. 6g), strongly suggesting that the lowered degradation of KIN11 in *naa20* (Fig. 6e) is caused by its lowered N-terminal acetylation status. In agreement with this claim, the non-acetylated KIN11^{D2P} was substantially stabilized in the wild type (Fig. 6f). To provide additional evidence for UPS-mediated degradation of KIN11, we performed cycloheximide chase assays in the presence and absence of the proteasome inhibitor MG132. In the absence of MG132, KIN11 was quickly degraded. Inhibition of the proteasome by MG132 entirely inhibited the degradation of KIN11 in the wild type. We have included these results in the revised manuscript (Lines 357-358, Supplementary Fig. 6j).

...if so, how does KIN11 N-terminal acetylation affect its ubiquitination and turnover?

Response: NTA of certain proteins has been reported to create specific signals for degradation via a particular branch of the ubiquitin proteasome system (UPS), termed the Ac/N-degron pathway, in which the Ac/N-degron is recognized by a specific E3 ubiquitin ligase termed N-recognin [16-19, reviewed in 20]. In our study, we demonstrated that KIN11 is a substrate of NatB both *in vivo* and *in vitro*. Our results indicate that N-terminally acetylated KIN11 is labelled for UPS-mediated degradation by an unidentified plant E3 ligase. This conclusion is further supported by showing that engineered KIN11 lacking N-terminal acetylation (KIN11^{D2P}) exhibits a reduced degradation rate in the wild type, suggesting that N-terminal acetylation of KIN11 promotes its degradation.

10. Multiple studies have shown that KIN10 is the major catalytic subunit of SnRK1 protein kinase complex, and *kin10* mutants are hypersensitive to abiotic stresses. Why does *kin10* not display stress-sensitive phenotypes in this study?

Response:

Indeed, KIN10 is suggested to be the major SnRK1 protein kinase in plants, because the KIN10 promoter is broadly active in all tissues, and the KIN11 promoter activity is highly active in generative organs [21]. Furthermore, immunoprecipitation assays using anti-SnRK1.1/SnRK1.2 antibodies revealed that complexes containing SnRK1.1 (KIN10) contribute approximately 90% of the total AMARA kinase activity. In comparison, complexes containing SnRK1.2 (KIN11) contribute only approximately 10% in crude extracts of *A. thaliana* cell cultures [22]. While using the same strategy and the same AMARA peptide, SnRK1 activity was assayed in the supernatant of seedling extracts after immunoprecipitation with antibodies to AKIN10 and AKIN11, and it was found that AKIN10 antibody removed 57.1% of SnRK1 activity, and AKIN11 antibody removed 38.3% of SnRK1 activity [23]. The latter finding suggests that KIN11 is more critical in Arabidopsis than previously thought. Moreover, based on the foundational research by Baena-González et al. and the latest transcriptomic analyses [24], KIN11 also exhibits very high expression levels in plants, if not comparable to those of KIN10 [25]. Consistent with this, both KIN10 and KIN11 single mutations display no phenotype under normal conditions, while double mutations are lethal, indicating functional redundancy between them [25].

Regarding the phenotype of the KIN10-depleted lines, depletion of KIN10 activity was reported to show no obvious phenotype after darkness, nitrogen starvation, drought, and submergence treatments [26]. In the initial study on SnRK1, Baena-González et al. demonstrated that soil-grown *kin10-1* and *kin10-2* lines did not show an obvious phenotype, whereas *kin10 kin11* VIGS double-mutant (d-1, d-2) plants arrest growth and exhibit premature senescence [25]. In another study, the *kin10kin11* sesqui mutant exhibited a dark-sensitive phenotype [27]. To the best of our knowledge, the *kin10* mutant did not show sensitivity under dark conditions, and the *kin11* single mutant has not been investigated previously. Ectopic overexpression of KIN10 confers resistance to drought stress [26], but a KIN11-overexpressing plant has not been tested in this study. Thus, we cannot conclude that this is a specific function of KIN10.

In our study, we investigated the resistance to darkness in *kin10* and *kin11* single mutants and found that only *kin11*, but not *kin10*, was more sensitive to darkness treatment (Supplementary Fig. 7a, b). To investigate whether the elevated KIN11 protein level is responsible for the increased darkness resistance of *naa20-cr1*, we crossed *naa20-cr1* with *kin10* and *kin11*. Remarkably, the darkness resistance of the *naa20-cr1 kin11* double mutant was attenuated compared with that of *naa20-cr1*. In contrast, inactivation of KIN10 in the *naa20-cr1 kin10* double mutant did not affect the darkness resistance of *naa20-cr1* (Fig. 7a, b). These results suggest that although the functions of KIN10 and KIN11 are largely redundant, the absence of KIN11 specifically impairs plants' resistance to darkness. Indeed, previous studies have also revealed functional differentiation between KIN10 and KIN11 in plants:

- (1) KIN10 is broadly expressed, while KIN11 expression is more pronounced in specific tissues and developmental stages [21].
- (2) While gain-of-function transgenic plants overexpressing two different isoforms of SnRK1.1 flower late, SnRK1.2 overexpressors flower early. In addition, SnRK1.2 overexpressors have increased leaf size and rosette diameter during early development, which is the opposite of SnRK1.1 overexpressors.

- (3) SnRK1 isoforms KIN10 and KIN11 are differentially regulated in Arabidopsis plants under phosphate starvation. Upon phosphate limitation, KIN11 is destabilized. The KIN10 protein remained stable [14].
- (4) KIN10, but not KIN11, can restore flowering and embryogenesis in the Arabidopsis trehalose 6-phosphate synthase1 (*tps1*) mutant [28].
- (5) In maize, only ZmSnRK α 2 (ZmKIN11), but not ZmSnRK α 1 nor ZmSnRK α 3, can interact with ZmWAK to regulate nutrient availability to defend against head smut disease [29].

11. What are the weak protein bands observed in the WT sample in Fig. S1B?

Response: The weak protein signals observed in the WT samples are endogenous plant proteins that are recognized by the commercial GFP antibody (ChromoTek, Pabg1-100). We have attached the uncropped WB results below and uploaded them to the “Source Data File”.

Response Fig. 3 Uncropped WB results of NAA20-GFP fusion protein in independent *pUB:NAA20-GFP/naa20-cr1* transgenic lines. Wild type (WT) was used as a control for the specificity of the GFP antibody.

12. Because proteasome activity is reduced in *naa20*, one would expect accumulation of ubiquitinated proteins under normal conditions. Why is this not observed?

Response: This assumption would be correct if the translation rate and the E3-ligase activity (defining the endogenous polyubiquitination rate) in *natb* mutants were not affected. We show in Fig. 2a that *natb* mutants exhibit lower extractable proteasome activity, but at the same time, they also exhibit a lower global translation rate (Fig. 2d). Since global degradation and global translation in the *natb* mutant are decreased, the steady-state proteome level (Fig. 2e) is almost unaffected, and the “global amount” of ubiquitinated proteins is not affected (Fig. 2c, compare WT DMSO with *naa20* DMSO). Furthermore, the accumulation of ubiquitinated proteins also depends on the endogenous E3 ligase activity. We claim that lowered NTA of NatB substrates causes less ubiquitination of NatB substrates. Even if the proteasome activity is decreased, polyubiquitinated proteins should only accumulate in *naa20*, if these proteins are translated and ubiquitinated at the same rate. Both translation (Fig. 2d) and global endogenous polyubiquitination rate (Fig. 2c, now compare the difference between DMSO and MG132 in the wild type versus the difference between DMSO and MG132 in *naa20*) are decreased in *naa20*. As a consequence, we do not observe a significant accumulation of polyubiquitinated in *naa20*.

13. Does NBR1 accumulate in the proteomics dataset of the *naa20* mutant? This should be reported explicitly.

Response: NBR1 has not been identified in the quantitative proteomics approach, likely due to its low protein abundance under normal conditions. This information is added in (Line 214).

14. Figures 4A–D show levels of autophagy-related proteins, but the treatment conditions are not specified. These must be clearly stated in the figure legends or main text.

Response: We apologize for not clearly reporting the exact treatment conditions. In this case, we immunologically detected proteins in the total protein fraction extracted from leaves of 6-week-old wild type, *naa20-cr1*, and *naa25-1* plants grown in soil under short-day conditions. We hope that the improved figure legend now contains all the necessary information to evaluate the obtained results.

15. Given the increased autophagy activity in *naa20*, do these plants exhibit delayed senescence, particularly under short-day conditions?

Response: We thank the expert reviewer for this helpful suggestion. As expected by the reviewer, the *natb* mutants exhibit delayed senescence after 10 weeks of growth in short-day conditions, followed by a 2-week transfer to long-day conditions. The senescent phenotype is consistent with the increased autophagy activity in *natb* mutants. We have added these novel data to Supplementary Fig. 5a and report them in the result section (Lines 289-291).

16. The statement “only three point 7 (3.788) proteins were identified” should be a typographical error.

Response: We thank the reviewer for carefully reading our manuscript and highlighting this error. We have corrected this statement (Line 168).

References

1. Jung, T.-Y., et al., *Naa20, the catalytic subunit of NatB complex, contributes to hepatocellular carcinoma by regulating the LKB1–AMPK–mTOR axis*. *Experimental & Molecular Medicine*, 2020. **52**: p. 1831–1844.
2. Emanuelle, S., et al., *SnRK1 from Arabidopsis thaliana is an atypical AMPK*. *Plant J*, 2015. **82**(2): p. 183–92.
3. Wurzinger, B., et al., *Redox state-dependent modulation of plant SnRK1 kinase activity differs from AMPK regulation in animals*. *FEBS Lett*, 2017. **591**(21): p. 3625–3636.
4. Crozet, P., et al., *Mechanisms of regulation of SNF1/AMPK/SnRK1 protein kinases*. *Frontiers in Plant Science*, 2014. **5**(190).
5. Shen, T., et al., *Function and molecular mechanism of N-terminal acetylation in autophagy*. *Cell Reports*, 2021. **37**(7): p. 109937.
6. Huber, M., et al., *NatB-Mediated N-Terminal Acetylation Affects Growth and Biotic Stress Responses*. *Plant Physiology*, 2020. **182**(2): p. 792–806.

7. Starheim, K.K., et al., *Identification of the human N(alpha)-acetyltransferase complex B (hNatB): a complex important for cell-cycle progression*. *Biochem J*, 2008. **415**(2): p. 325–31.
8. Plevoda, B., et al., *Nat3p and Mdm20p are required for function of yeast NatB Nalpha-terminal acetyltransferase and of actin and tropomyosin*. *J Biol Chem*, 2003. **278**(33): p. 30686–97.
9. Linster, E., et al., *Cotranslational N-degron masking by acetylation promotes proteome stability in plants*. *Nature Communications*, 2022. **13**(1): p. 810.
10. Ji, C., et al., *AtNBR1 Is a Selective Autophagic Receptor for AtExo70E2 in Arabidopsis*. *Plant Physiol*, 2020. **184**(2): p. 777–791.
11. Klionsky, D.J., et al., *Guidelines for the use and interpretation of assays for monitoring autophagy (4th edition)(1)*. *Autophagy*, 2021. **17**(1): p. 1–382.
12. Qi, H., et al., *Studying plant autophagy: challenges and recommended methodologies*. *Advanced Biotechnology*, 2023. **1**(4): p. 2.
13. Varshavsky, A., *N-degron pathways*. *Proceedings of the National Academy of Sciences*, 2024. **121**(39): p. e2408697121.
14. Frago, S., et al., *SnRK1 Isoforms AKIN10 and AKIN11 Are Differentially Regulated in Arabidopsis Plants under Phosphate Starvation*. *Plant Physiology*, 2009. **149**(4): p. 1906–1916.
15. Bachmair, A., D. Finley, and A. Varshavsky, *In vivo half-life of a protein is a function of its amino-terminal residue*. *Science*, 1986. **234**(4773): p. 179–86.
16. Heathcote, K.C., et al., *N-terminal cysteine acetylation and oxidation patterns may define protein stability*. *Nature Communications*, 2024. **15**(1): p. 5360.
17. Varland, S., et al., *N-terminal acetylation shields proteins from degradation and promotes age-dependent motility and longevity*. *Nat Commun*, 2023. **14**(1): p. 6774.
18. Hwang, C.S., A. Shemorry, and A. Varshavsky, *N-terminal acetylation of cellular proteins creates specific degradation signals*. *Science*, 2010. **327**(5968): p. 973–7.
19. Shemorry, A., C.S. Hwang, and A. Varshavsky, *Control of protein quality and stoichiometries by N-terminal acetylation and the N-end rule pathway*. *Mol Cell*, 2013. **50**(4): p. 540–51.
20. Arnesen, T., I. Kjosås, and N. McTiernan, *Protein N-terminal acetylation is entering the degradation end game*. *Nature Reviews Molecular Cell Biology*, 2024. **25**: p. 335–336.
21. Williams, S.P., et al., *Regulation of Sucrose non-Fermenting Related Kinase 1 genes in Arabidopsis thaliana*. *Front Plant Sci*, 2014. **5**: p. 324.
22. Jossier, M., et al., *SnRK1 (SNF1-related kinase 1) has a central role in sugar and ABA signalling in Arabidopsis thaliana*. *Plant J*, 2009. **59**(2): p. 316–28.
23. Zhang, Y., et al., *Inhibition of SNF1-related protein kinase1 activity and regulation of metabolic pathways by trehalose-6-phosphate*. *Plant Physiol*, 2009. **149**(4): p. 1860–71.
24. Xu, Q., F. Kong, and W. Yang, *SnRK1 as the Core Node Integrating Energy Homeostasis, Stress Adaptation and Hormonal Crosstalk in Plants*. *Plant Cell Environ*, 2025. **48**(11): p. 7830–7847.
25. Baena-Gonzalez, E., et al., *A central integrator of transcription networks in plant stress and energy signalling*. *Nature*, 2007. **448**(7156): p. 938–42.
26. Chen, L., et al., *The AMP-Activated Protein Kinase KIN10 Is Involved in the Regulation of Autophagy in Arabidopsis*. *Frontiers in Plant Science*, 2017. **Volume 8 - 2017**.
27. Pedrotti, L., et al., *Snf1-RELATED KINASE1-Controlled C/S1-bZIP Signaling Activates Alternative Mitochondrial Metabolic Pathways to Ensure Plant Survival in Extended Darkness*. *The Plant Cell*, 2018. **30**(2): p. 495–509.
28. Zacharaki, V., et al., *Impaired KIN10 function restores developmental defects in the Arabidopsis trehalose 6-phosphate synthase1 (tps1) mutant*. *New Phytol*, 2022. **235**(1): p. 220–233.
29. Zhang, Q., et al., *A maize WAK-SnRK1α2-WRKY module regulates nutrient availability to defend against head smut disease*. *Molecular Plant*, 2024. **17**(11): p. 1654–1671.

REVIEWER COMMENTS

Reviewer #1 (Remarks to the Author):

I am satisfied that the authors have addressed my comments. They have also provided several pieces of additional data to address the comments of the other reviewers.

Response: We greatly appreciate the reviewer's valuable evaluation and constructive suggestions, which have significantly improved the manuscript.

Reviewer #2 (Remarks to the Author):

I am satisfied with the revised version of the manuscript submitted to Nature Communications. The authors have adequately addressed all of my queries and comments.

Response: We greatly appreciate the reviewer's valuable evaluation and constructive suggestions, which have significantly improved the manuscript.

Reviewer #3 (Remarks to the Author):

The authors have made an effort to address the previous concerns. However, several critical issues remain regarding the interpretation of the autophagy data and the genetic evidence supporting the KIN11 pathway. The following points should be addressed to improve the quality of the manuscript:

1. The authors conclude that autophagy is upregulated in the *naa20* mutant based on increased ATG8 lipidation, ATG5-ATG12 conjugation, and the presence of autophagic bodies. However, these markers can be misinterpreted if autophagic flux (the complete degradation process) is not properly distinguished from the mere accumulation of autophagosomes.

(1) ATG8 Lipidation: Increased levels of ATG8-PE are often observed in mutants where autophagosome-vacuole fusion is disrupted (reduced flux), not just when autophagy is induced. Furthermore, the protein bands in Figure 4c are ambiguous; the lower faint band may not represent the lipidated form. The authors must include an *atg5* or *atg7* mutant (deficient in ATG8-PE formation) as a negative control to definitively identify the ATG8-PE band.

Response: We thank the expert reviewer for the critical interpretation of autophagic flux assessment and for providing detailed revision suggestions. As suggested, we added an *atg5-1* mutant as a negative control to provide additional evidence for the identity of the ATG8-PE signal. Absence of the proposed ATG8-PE signal in *atg5-1* mutant clearly demonstrates that this signal indeed corresponds to ATG8-PE, which accumulates in the *natb* mutants compared with the wild type (Fig. 4c, lines 263-264, also see Response Fig. 1). In this experiment, total proteins were extracted using NuPage sample buffer (150 mM Tris-HCl, pH 8.5, 2% LDS, 1 mM EDTA, 10% [v/v] glycerol, 0.22 mM Coomassie Blue G250, 0.166 mM Phenol Red, and 50 mM DTT) and subsequently separated by Tricine-SDS-PAGE (Schägger, *Nature protocols*, 2006), which enables us to distinguish the different ATG8 variants.

Response Fig. 1 Immunoblot detection of ATG8 from the leaves of 6-week-old wild type (WT) and *natb* mutants grown in soil under short-day conditions. The *atg5-1* mutant was used as a control for antibody specificity. The red arrow indicates ATG8-PE form. Protein levels were normalized to the loading control (LC) and the detected signal was set to one in the WT.

References:

Schägger, H. (2006). Tricine–sds-page. *Nature protocols*, 1(1), 16-22.

(2) ATG5-ATG12 Conjugation: In plant systems, it is established that most endogenous ATG5 exists in the conjugated (ATG5-ATG12) form (see PMC3967041). In Figure 4e, the majority of ATG5 appears as a free protein. This contradicts established literature and requires a technical explanation or re-validation.

Response: We thank the reviewer for pointing out the potentially misleading interpretation of the Western blot. The reviewer is correct that most endogenous ATG5 exists in the conjugated ATG5–ATG12 form. This signal is indicated by a red arrow in the figure. The lower signal shown in the uncropped version of the previous manuscript was nonspecific (see Response Fig. 2A) and was therefore not labelled as free ATG5 in the resubmitted manuscript (see Response Fig. 2B -> old Figure 4E). To avoid misinterpretation, we will show only the specific signal for the ATG5-ATG12 complex in the revised version of the manuscript (revised Fig. 4e; included here as Response Fig. 2C). We did not detect any signal for free ATG5 with the ATG5 antibody (Agrisera, AS15 3060) used here.

Response Fig. 2 Immunoblot detection of ATG5-ATG12 from the leaves of 6-week-old wild type (WT) and *natb* mutants grown in soil under short-day conditions. The *atg5-1* mutant was used as a control for antibody specificity, but not shown in the cropped version of the western blot. (A) Uncropped immunoblot and amido black staining of the membrane. The red arrow indicates the absence of the ATG5-ATG12 complex in the *atg5-1* mutant. (B) Fig. 4e shown in the previous manuscript. The red arrow indicates the ATG5-ATG12 complex. The lower band is a

nonspecific signal detected by the ATG5 antibody (Agriser, AS15 3060). (C) Improved Fig. 4e shown in the revised manuscript. The red arrow indicates ATG5-ATG12 complex. In (B and C), protein levels were normalized to the loading control (LC) and the detected signal was set to one in the WT.

(3) Confocal Imaging and Vacuolar Localization: The "dots" identified as autophagic bodies in Figure 4g appear to be localized in the cytoplasm rather than within the vacuole. If these structures are cytoplasmic, it suggests a block in trafficking/fusion, implying decreased rather than increased flux. The authors should redo confocal analysis using root elongation cells treated with Concanamycin A to stabilize autophagic bodies within the vacuole, and take images to demonstrate the vacuolar autophagic bodies following the procedures shown in some previous publications (e.g., Zhao et al., *J Cell Biol*, 2022; Stephani et al., *eLife*, 2020). ...

Response: We appreciate the reviewer's careful assessment and detailed experimental instructions. We have repeated the confocal analysis following the procedures described in Yang et al., *Mol. Plant*, 2023, Zhao et al., *J Cell Biol*, 2022 and Stephani et al., *eLife*, 2020. The confocal images clearly showed that after Concanamycin treatment, autophagic bodies appeared in the vacuole of elongated root cells in the wild type and the *naa20-cr1* plants (Fig. 4g, also see Response Fig. 3).

Response Fig. 3 Confocal microscopy analysis of the autophagic flux in root elongation cells. Five-day-old *pUBQ::GFP-ATG8a/WT* and *pUBQ::GFP-ATG8a/naa20-cr1* seedlings were grown on ½ MS medium plates containing 1% sucrose under long day conditions and exposed to continuous light for 1 day before being transferred to liquid ½ MS medium supplemented with or without 1 μM ConA followed by incubation in the dark for 6 hours. The

formation of autophagic bodies in cells of the root elongation zones was detected. A representative picture from 10 individual seedlings is shown. Scale bar, 50 μm . Inset scale bars, 10 μm .

References:

1. Yang, C., Li, X., Yang, L., Chen, S., Liao, J., Li, K., ... & Gao, C. (2023). A positive feedback regulation of SnRK1 signaling by autophagy in plants. *Molecular plant*, 16(7), 1192-1211.
2. Zhao, J., Bui, M. T., Ma, J., Künzl, F., Picchianti, L., De La Concepcion, J. C., ... & Dagdas, Y. (2022). Plant autophagosomes mature into amphisomes prior to their delivery to the central vacuole. *Journal of Cell Biology*, 221(12), e202203139.
3. Stephani, M., Picchianti, L., Gajic, A., Beveridge, R., Skarwan, E., Sanchez de Medina Hernandez, V., ... & Dagdas, Y. (2020). A cross-kingdom conserved ER-phagy receptor maintains endoplasmic reticulum homeostasis during stress. *elife*, 9, e58396.

... Moreover, to provide definitive proof of autophagic flux, a standard GFP-cleavage immunoblotting assay should be performed.

Response: As suggested, we have performed a standard GFP-cleavage assay to assess the impact of darkness stress on the autophagic flux in the wild type and the *naa20-cr1* mutant. To this end, we suppressed autophagy as much as possible in both genotypes by growing seedlings on 1/2 MS medium plates containing 1% sucrose under long day conditions for five days, followed by 24 hours of continuous light (time point 0). Next, the wild type and *naa20-cr1* mutant were subjected to darkness for up to 12 hours as described in Yang et al., *Mol. Plant*, 2023. Darkness treatment caused a substantially stronger induction of GFP release from GFP-ATG8a in the *naa20-cr1* when compared with the wild type as indicated by the higher GFP to GFP-ATG8a ratio in *naa20-cr1* at all times of darkness treatment (4, 8, and 12 hours) (see Response Fig. 4). These novel results are included in Fig. 4i and reported in Lines 271-276.

Response Fig. 4 GFP-ATG8a cleavage assay showing a higher GFP/GFP-ATG8a ratio in the *naa20-cr1* background upon carbon-starvation treatment. Five-day-old seedlings grown on 1/2 MS medium plates containing 1% sucrose under long day conditions were exposed to continuous light for 1day before being subjected to liquid 1/2 MS-C medium and kept in dark for 0, 4, 8, and 12 h. Then the seedlings were harvested to isolate total proteins for immunoblotting analysis using anti-GFP antibody. The values below the immunoblot are the relative ratios of free GFP to GFP-ATG8a. Amido black staining of proteins transferred to the PVDF membrane served as loading control.

References:

1. Yang, C., Li, X., Yang, L., Chen, S., Liao, J., Li, K., ... & Gao, C. (2023). A positive feedback regulation of SnRK1 signaling by autophagy in plants. *Molecular plant*, 16(7), 1192-1211.

2. The authors claim that the *kin11* single mutant is sensitive to dark treatment and posits that the dark-tolerant phenotype of *naa20* is due to elevated KIN11 (but not KIN10) activity. To strengthen this causal link, the authors should provide evidence from gain-of-function lines. Have the authors tested the dark-tolerance phenotypes of KIN11-OE or KIN11(D2P)-OE (phosphomimetic) overexpression plants?

Response: As suggested by the referee, we have tested the darkness tolerance of two independent KIN11-overexpressing lines. As expected, overexpression of KIN11 in the wild type (KIN11-OE) resulted in higher survival rates of darkness-treated KIN11-OE plants when compared to wild type (see Response Fig. 5). This increased survival rate was comparable to that of the *naa20-cr1* mutant upon darkness treatment. These novel results are included in Supplementary Fig. 7c, d, and are reported in Lines 379-382.

Response Fig. 5 Representative images and survival rates of wild type (WT), *naa20-cr1*, *kin11* and two independent *KIN11-GFP* overexpressing lines after 10-day carbon starvation and recovery. Data are shown as means \pm SEM, $n = 3$, every 40 plants were grouped as one set. Circles indicate individual data points. Different letters indicate individual groups identified by pairwise multiple comparisons with a one-way ANOVA followed by a Tukey's test ($p < 0.05$). Scale bar, 2 cm.